# Agentic Jigsaw Interaction Learning for Enhancing Visual Perception and Reasoning in Vision-Language Models

**Yu Zeng**[1*], **Wenxuan Huang**[3,4*], **Shiting Huang**[1*], **Xikun Bao**[1], **Yukun Qi**[1], **Yiming Zhao**[1],

**Qiuchen Wang**[1], **Lin Chen**[1], **Zehui Chen**[1], **Huaian Chen**[1], **Wanli Ouyang**[2,4], **Feng Zhao**[1†]

[1]University of Science and TechnolZengogy of China
[2]Shanghai AI Laboratory
[3]East China Normal University
[4]The Chinese University of Hong Kong

## Abstract

Although current large Vision-Language Models (VLMs) have advanced in multimodal understanding and reasoning, their fundamental perceptual and reasoning abilities remain limited. Specifically, even on simple jigsaw tasks, existing VLMs perform near randomly, revealing deficiencies in core perception and reasoning capabilities. While high-quality vision-language data can enhance these capabilities, its scarcity and limited scalability impose significant constraints. To address this, we propose **AGILE**, an **A**gentic ji**G**saw **I**nteraction **L**earning for **E**nhancing visual perception and reasoning in VLMs. AGILE formulates jigsaw solving as an interactive process, enabling the model to progressively engage with the environment. At each step, the model generates executable code to perform an action based on the current state, while the environment provides fine-grained visual feedback to guide task completion. Through this iterative cycle of observation and interaction, the model incrementally improves its perceptual and reasoning capabilities via exploration and feedback. Experimental results show that AGILE not only substantially boosts performance on jigsaw tasks of varying complexity (e.g., increasing accuracy from 9.5% to 82.8% under the $2 \times 2$ setting) but also demonstrates strong generalization across 9 general vision tasks, achieving an average improvement of 3.1%. These results indicate notable enhancements in both perceptual and reasoning abilities. This work opens a new avenue for advancing reasoning and generalization in multimodal models and provides an efficient, scalable solution to the scarcity of multimodal reinforcement learning data. The code and datasets is available at https://github.com/yuzeng0-0/AGILE.

## 1 Introduction

Large Vision-Language Models (VLMs) have recently achieved remarkable success across a wide range of multimodal tasks, including image captioning (Chen et al., 2024a;c; Li et al., 2024b; Zeng et al., 2025), visual question answering (Bai et al., 2025; Zhu et al., 2025; Hurst et al., 2024; Comanici et al., 2025; Qi et al., 2025; Zhao et al., 2025; Huang et al., 2026b; Zeng et al., 2026; Wang et al., 2026; Pan et al., 2025), document understanding (Wang et al., 2025b;a), and complex real-world applications (Huang et al., 2025b; Han et al., 2026; Fang et al., 2025; Huang et al., 2025a; Zhang et al., 2025a). By effectively associating visual and textual information, these models exhibit strong multimodal perception and reasoning capabilities. However, their performance remains severely limited on tasks that require comprehensive visual understanding and structured reasoning. In particular, we find that current VLMs often perform near random even on relatively simple $2 \times 2$ jigsaw tasks (Carlucci et al., 2019; Chen et al., 2023), which demand both perceptual accuracy and

---

\*    Equal Contribution
†    Corresponding author

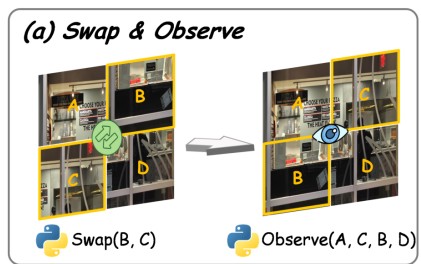 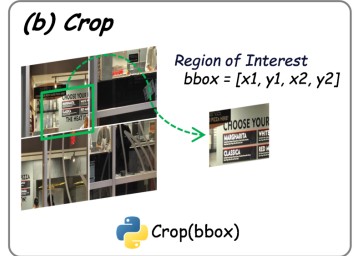 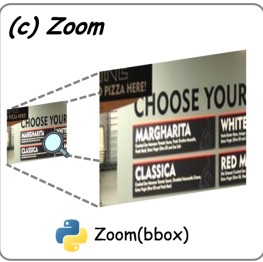

Figure 1: **Description of the action space.** (a) illustrates swapping two jigsaw pieces and observing the updated jigsaw state; (b) shows cropping a specific region of the jigsaw for closer inspection; and (c) depicts zooming into a selected area to examine fine-grained details.

logical inference. This suggests that existing pretraining and fine-tuning strategies are insufficient for developing robust perceptual and reasoning abilities.

Recent studies have explored reinforcement learning (RL) (Schulman et al., 2017; Rafailov et al., 2023; Shao et al., 2024; Huang et al., 2026a; Ma et al., 2025; Zhang et al., 2025c;b; Ren et al., 2026) as a way to enhance these capabilities by enabling models to learn through interaction, trial-and-error, and feedback. While RL-based approaches show promise, they are fundamentally constrained by the scarcity and limited scalability of high-quality vision-language RL data. Current methods for constructing multimodal RL datasets (Huang et al., 2025c; Yang et al., 2025; Lu et al., 2023) generally fall into two categories: human expert supervision and automated synthesis. Human-supervised datasets are either prohibitively expensive or too small in scale for large-scale training, while automated approaches that rely on closed-source models suffer from limited quality, capability constraints, and substantial API costs. These challenges collectively hinder the development of VLMs with strong reasoning and generalization abilities.

To address these limitations, we propose **AGILE**, an **A**gentic ji**G**saw **I**nteraction **L**earning framework for **E**nhancing visual perception and reasoning in VLMs. Our approach leverages the structured nature of the jigsaw puzzle as a proxy task for perception and reasoning, modeling the jigsaw-solving process as a progressive interaction between the model and its environment. At each step, the model generates Python code to perform actions within a well-defined action space (swapping two image tiles, observing the current jigsaw state, cropping regions for detailed observation, or zooming in for fine-grained analysis). This interaction-driven process enables the model to iteratively refine its perceptual and reasoning capabilities while receiving explicit feedback at every step. By simulating the step-by-step dynamics of jigsaw solving, the model is encouraged to capture structural relationships among visual components and acquire more robust perception and reasoning skills.

A key feature of our approach is the use of code and rule-based data generation, which offers two major advantages. First, the difficulty of the jigsaw task can be precisely controlled by adjusting factors such as the number of correctly placed tiles in the initial state and the overall jigsaw size. Second, because the ground-truth solution is inherently available, the synthetic dataset can be scaled to arbitrary sizes with strict supervision, overcoming the data scarcity inherent in human-annotated or closed-source datasets. The combination of interactive training and scalable data generation yields an efficient and effective framework for advancing visual perception and reasoning in VLMs.

We validate the effectiveness of our approach through comprehensive experiments. Our method substantially improves performance across jigsaw tasks of varying complexity, for instance, raising accuracy from 9.5% to 82.8% on the $2 \times 2$ setting, while also demonstrating strong generalization to a wide range of vision tasks, including high-resolution image understanding, real-world scene analysis, fine-grained recognition, visual reasoning, and hallucination benchmarks. Moreover, we show that scaling the training data further boosts performance, and that under equal data budgets, jigsaw-based training achieves results comparable to or even surpassing those obtained with general QA data. These findings underscore the potential of jigsaw tasks in addressing the scarcity of multimodal RL data. By combining interactive training with scalable data generation, our method significantly enhances both perceptual and higher-order reasoning abilities, pointing to a promising new direction for advancing VLMs. Our main contributions are as follows:

- We introduce AGILE, an agentic jigsaw interaction learning framework that formulates jigsaw solving as a stepwise interactive process, thereby driving incremental improvements in both visual perception and reasoning capabilities of VLMs.

- We present a scalable jigsaw-based data generation method that yields high-quality multimodal reinforcement learning datasets with controllable difficulty, providing an efficient solution to the current shortage of high-quality training data.

- Extensive experiments demonstrate that our approach substantially improves performance on jigsaw tasks of varying complexity while exhibiting strong generalization across diverse vision benchmarks, highlighting its effectiveness in enhancing both perception and reasoning in VLMs.

## 2 RELATED WORK

**Reinforcement Learning for Vision Language Models.** Reinforcement learning (RL) (Schulman et al., 2017; Rafailov et al., 2023; Shao et al., 2024) has been widely applied to improve reasoning in language models, as exemplified by the success of DeepSeek-R1 (Guo et al., 2025) in mathematical reasoning. Building on this progress, recent studies have extended RL to VLMs, with rule-based RL in multimodal domains emerging as a promising direction. For perception enhancement, R1-V (Chen et al., 2025b) applies RL to object counting, while Perception-R1 (Yu et al., 2025) leverages object matching and IoU as reward signals to improve grounding. DeepEyes (Zheng et al., 2025) shows how RL can encourage models to invoke visual tools, thereby expanding perceptual abilities. For reasoning, MMEureka (Meng et al., 2025) demonstrates the effectiveness of rule-based RL in mathematical tasks. From a data perspective, Vision-R1 (Huang et al., 2025c) and R1-OneVision (Yang et al., 2025) convert visual information into textual representations to build multimodal chain-of-thought (CoT) datasets that support stronger reasoning. Despite these advances, the lack of scalable, high-quality RL datasets remains a fundamental bottleneck, severely limiting further improvements in both perception and reasoning for VLMs.

**Enhancing Generalization via Proxy Tasks.** Rule-based reinforcement learning (RL) has shown promise but typically requires large amounts of high-quality, verifiable data. Logic-RL (Xie et al., 2025a) addressed this by introducing the "Knights and Knaves" (K&K) puzzle, which offers controllable difficulty and algorithmically generated ground truth, enabling models to acquire reasoning skills transferable to mathematical tasks. Enigmata (Chen et al., 2025a) extended this approach to a broader set of text-based puzzles (e.g., cryptographic, arithmetic, logical), demonstrating that solving such tasks with RL improves general reasoning without external data. RPT (Dong et al., 2025) further reframed next-token prediction as a reasoning task, using verifiable rewards to enhance predictive ability and strengthen the basis for reinforcement fine-tuning. Inspired by these proxy-task successes in LLMs, recent work has begun exploring similar ideas for VLMs. Code2Logic (Tong et al., 2025) and ViGaL (Xie et al., 2025b) leverage code-synthesized games to improve mathematical reasoning, while ViCrit (Wang et al., 2025d) enhances perceptual robustness by reinforcing models to detect hallucinated entities via modified captions. Jigsaw-R1 (Wang et al., 2025e) proposed a puzzle-based RL paradigm, but due to training limitations, its models still performed poorly even on $2 \times 2$ jigsaw puzzles, failing to fully exploit the proxy-task benefits. To address this gap, we propose a agentic jigsaw interaction learning framework, which casts jigsaw solving as an interactive process between the model and environment. This enables iterative refinement of perception and reasoning with explicit stepwise feedback, allowing the model to better capture visual relationships and develop stronger perceptual and reasoning skills.

## 3 METHOD

In this section, we provide a comprehensive description of our proposed agentic jigsaw interaction learning framework. We first introduce the jigsaw task and describe how the model interacts with the environment to accomplish it (Sec. 3.1), then present the construction of the training data (Sec. 3.2). Finally, we outline the training paradigm, including both the cold-start stage and the reinforcement learning stage (Sec. 3.3).

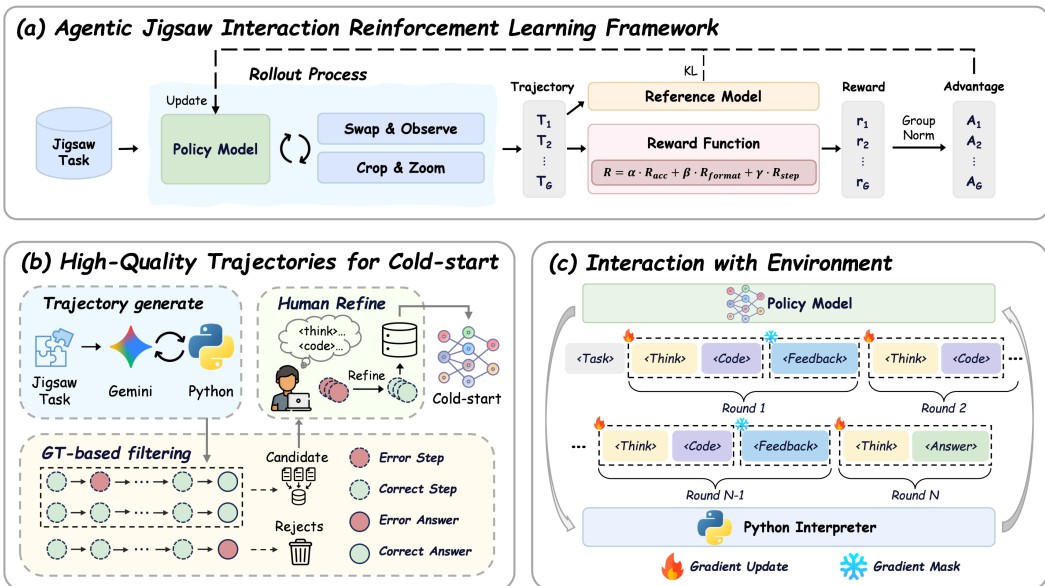

Figure 2: **Overview of the AGILE framework.** (a) depicts the interaction process between the model and the external environment, together with the implementation of the GRPO algorithm; (b) shows the collection of high-quality jigsaw trajectory data; and (c) illustrates the model–environment interaction during the jigsaw rollout process.

## 3.1 JIGSAW TASK AND ENVIRONMENT INTERACTION

**Jigsaw Task.** Given an input image, we partition it into an $m \times m$ grid of jigsaw pieces, where task difficulty can be flexibly adjusted by varying $m$. If the image height or width is not divisible by $m$, the image is resized so that its dimensions are exact multiples of the grid. The grid is then randomly shuffled, and each piece is assigned an index in row-major order ranging from 1 (top-left) to $m^2$ (bottom-right). The shuffled configuration is denoted as

$$I_{Shuffle} = \{I_1, I_2, \ldots, I_{m^2}\}, \tag{1}$$

where each $I_k$ corresponds to one jigsaw block. Since the shuffling process is explicitly recorded during data generation, we can recover the ground-truth jigsaw layout as

$$I_{GT} = \{I_{\pi(1)}, I_{\pi(2)}, \ldots, I_{\pi(m^2)}\}, \tag{2}$$

where $\pi$ is a permutation over $\{1, 2, \ldots, m^2\}$. During jigsaw solving, the model will maintain a current state

$$I_{State} = \{I_{\pi^*(1)}, I_{\pi^*(2)}, \ldots, I_{\pi^*(m^2)}\}, \tag{3}$$

where $\pi^*$ denotes the current arrangement of pieces. At each step, the model iteratively swaps two pieces, observes the resulting configuration, and aims to reconstruct the ground-truth layout $I_{GT}$.

**Environment Interaction.** The model interacts with the environment throughout the jigsaw-solving process in an iterative manner. As illustrated in Figure 1, , we predefine a set of API for model–environment interactions in Python. The model expresses its actions by generating Python code, which is executed by the environment to produce the corresponding resulting image. The model then uses these results to reason further and decide on subsequent actions, repeating this process iteratively until the jigsaw is completed. Specifically, at each step, the model can perform the following actions:

• **Swap:** Given the current jigsaw state $I_{State}$, the model calls the `Swap` function to exchange the positions of any two jigsaw pieces.

• **Observe:** Given $I_{State}$, the model calls the `Observe` function to obtain the current jigsaw progress $I_{Obs}$, which is then used to determine the next action.

• **Crop & Zoom:** Given $I_{Obs}$, the model crops and zooms into a local region to inspect finer details and inform subsequent actions.

To better illustrate the overall procedure, the entire rollout process is presented in Algorithm 1.

---

**Algorithm 1** Interactive Jigsaw-Solving Procedure of VLMs with the Environment

---

**Input:** Query $x$, $I_{Shuffle}$, Policy model $\pi_\theta$, External environment $\mathcal{V}$, Maximum iteration steps $T$.
**Output:** Final interaction trajectory $y$.
 1: Initialize trajectory $y \leftarrow \emptyset$, step counter $t \leftarrow 0$.
 2: **while** $t < T$ **do**
 3:     Sample response $y_t \sim \pi_\theta(\cdot \mid x, I_{Shuffle}, y)$.
 4:     Append assistant response $y_t$ to trajectory: $y \leftarrow y + y_t$.
 5:     **if** ` ` detected in $y_t$ **then**
 6:         Parse the Python code (e.g., Swap, Observe, Crop, Zoom), execute it in the environment $\mathcal{V}$, and obtain feedback $O_t$.
 7:     **else if** `<answer> </answer>` detected in $y_t$ **then**
 8:         **return** final trajectory $y$.
 9:     **end if**
10:     Append environment feedback $O_t$ as user input: $y \leftarrow y + O_t$.
11:     Update step counter $t \leftarrow t + 1$.
12: **end while**
13: **return** final trajectory $y$.

---

### 3.2 Data Construction

In Section 3.1, we have introduced the jigsaw task and described how the model interacts with the environment to complete it. Based on this task design, we observe that the model (e.g., Qwen-2.5VL-7B) exhibits limited foundational capabilities, manifested in poor instruction-following and incorrect Python code generation, which hinders proper interaction with the environment and introduces substantial training noise. Directly applying reinforcement learning (RL) thus results in low learning efficiency. To bridge this gap and more effectively train our model, we employ Gemini 2.5 Pro[1] to collect expert trajectories for the cold-start phase, endowing Qwen-2.5VL-7B with basic interactive jigsaw-solving capabilities. For this purpose, we design structured prompts to guide Gemini 2.5 Pro in interacting with the environment to complete the jigsaw tasks (The prompt templates are provided in Appendix B. ).

To further ensure data quality, we apply an additional filtering process: first, selecting samples where Gemini's outputs match the ground truth, and then manually verifying each interaction step in the correct samples to ensure rationality and consistency. To ensure that our model can perform diverse jigsaw reasoning actions during reinforcement learning, we carefully balance the training data. Specifically, trajectories are balanced with respect to both the number of steps (4–8) and the types of actions involved (e.g., Swap, Observe, Crop, Zoom). This design ensures that the model is sufficiently exposed to the full action space and learns to handle various interactions with the environment effectively. Overall, we curate a dataset comprising 1.6K high-quality reasoning trajectories for the jigsaw task.

### 3.3 Training Paradigm: Cold-start and Reinforcement Learning

**Cold-start.** During the cold-start phase, we leverage the 1.6K high-quality jigsaw-solving trajectories constructed in Section 3.2 to equip the model with basic instruction-following and Python code generation capabilities, ensuring effective interaction with the environment.

**Reinforcement Learning.** Our framework adopts Group Relative Policy Optimization (GRPO), which utilizes the average reward of multiple sampled outputs as a baseline, instead of relying on a learned value function. The policy model is optimized by maximizing the following objective:

---

[1]The version is `Gemini-2.5-Pro-Preview-05-06`.

$$\mathcal{J}_{\mathrm{GRPO}}(\theta) = \mathbb{E}_{x \sim \mathcal{D}, \, \{y_i\}_{i=1}^{G} \sim \pi_{\mathrm{old}}(\cdot | x; \mathcal{V})} \left[ \frac{1}{G} \sum_{i=1}^{G} \frac{1}{\sum_{t=1}^{|y_i|} I(y_{i,t})} \sum_{t=1:I(y_{i,t})=1}^{|y_i|} \min \left( \frac{\pi_\theta(y_{i,t} \mid x, y_{i,<t}; \mathcal{V})}{\pi_{\mathrm{old}}(y_{i,t} \mid x, y_{i,<t}; \mathcal{V})} \hat{A}_{i,t}, \right. \right.$$
$$\left. \left. \mathrm{clip} \left( \frac{\pi_\theta(y_{i,t} \mid x, y_{i,<t}; \mathcal{V})}{\pi_{\mathrm{old}}(y_{i,t} \mid x, y_{i,<t}; \mathcal{V})}, 1-\epsilon, 1+\epsilon \right) \hat{A}_{i,t} \right) \right] - \beta \, \mathbb{D}_{\mathrm{KL}}(\pi_\theta \, \| \, \pi_{\mathrm{ref}}),$$

$$(4)$$

where the rollout module samples a group of trajectories $\{y_1, y_2, \ldots, y_G\}$ from the old policy $\pi_{\mathrm{old}}$ for each input question $x$ through interaction with the external environment $\mathcal{V}$. The advantage term $\hat{A}_{i,t}$ is computed based on the relative rewards of outputs within each group. Our reward system comprises three components: an accuracy reward, a format reward, and a step reward. The total reward is computed as the sum of these components.

• **Accuracy Reward:** We compare the model's generated answer $I_{Answer}$ with the ground truth $I_{GT}$. If all jigsaw image blocks are correctly placed, the accuracy reward is 1; otherwise, it is 0.

• **Format Reward:** The model receives a reward of 1 if its output follows the required structured format, with the reasoning process, code, and final answer correctly enclosed within the `<think>`, ``, and `<answer>` tags, respectively.

• **Step Reward:** We encourage the model to complete the jigsaw in as few steps as possible. For a $2 \times 2$ jigsaw, if each step swaps any two image blocks, at most three steps are theoretically required to place all blocks correctly. Excessive actions may lead the model to perform invalid steps, reducing the proportion of effective perception and reasoning. Furthermore, to prevent the model from prematurely hacking the step reward during early RL training, the step reward is applied only when the jigsaw is correctly completed. If the jigsaw is incorrect, the model is penalized by assigning the maximum step penalty. Formally, the step reward is defined as:

$$R_{\mathrm{step}} = \lambda \cdot \left( \mathbb{I}_{\{R_{\mathrm{acc}}=1\}} \cdot step_{\mathrm{num}} + \mathbb{I}_{\{R_{\mathrm{acc}}=0\}} \cdot step_{\mathrm{max}} \right), \tag{5}$$

where $step_{\mathrm{num}}$ is the number of steps the model actually used to complete the jigsaw, $step_{\mathrm{max}}$ is the maximum allowed steps, $\lambda$ denotes the step penalty coefficient, which is set to $-0.05$, and $\mathbb{I}_{\{\cdot\}}$ is the indicator function. The final reward formulation is shown in Equation 6:

$$R = \alpha \cdot R_{\mathrm{acc}} + \beta \cdot R_{\mathrm{format}} + \gamma \cdot R_{\mathrm{step}}, \tag{6}$$

where the coefficients $\alpha$, $\beta$, and $\gamma$ weight the relative importance of accuracy, format, and step rewards, respectively. In our experimental setup, $\alpha$, $\beta$, and $\gamma$ are set to 0.8, 0.2, and 1.0, respectively.

## 4 EXPERIMENT

In this section, we present a comprehensive experimental study. We first describe the implementation details, including datasets, training procedures, and inference settings (Sec. 4.1). We then evaluate the effectiveness of our agentic jigsaw interaction learning framework on both jigsaw-solving and general vision tasks (Sec. 4.2). Finally, we perform ablation studies to assess the impact of jigsaw data scale and to contrast the benefits of jigsaw data with those of general QA data (Sec. 4.3).

### 4.1 IMPLEMENTATION DETAILS

**Datasets and Models.** Our training data consist of two components corresponding to the cold-start and reinforcement learning (RL) stages. In the cold-start stage, we employ 1.6K high-quality jigsaw puzzle trajectories collected in Sec. 3.2 to endow the model with basic interactive jigsaw-solving skills. In the RL stage, we construct a dataset of 15.6K images spanning diverse domains, including high-resolution visual search, OCR-based text recognition, real-world scenes, and structured diagrams (see Appendix A for details). Each image is partitioned into $2 \times 2$ patches and randomly shuffled to ensure that all patches are initially misplaced. All experiments are conducted using Qwen2.5-VL-7B (Bai et al., 2025) as the base model.

Table 1: **Jigsaw Acc result.** LN indicates the difficulty level, where N denotes the initial number of correct pieces. A smaller N corresponds to a more scrambled jigsaw and higher difficulty. The best results are highlighted in **bold**, and the second-best results are underlined.

| Model | $2 \times 2$ | | | | $3 \times 3$ | | | | | | | | |
|---|---|---|---|---|---|---|---|---|---|---|---|---|---|
| | L0 | L1 | L2 | Avg. | L0 | L1 | L2 | L3 | L4 | L5 | L6 | L7 | Avg. |
| Random | 4.5 | 3.7 | 4.2 | 4.1 | 0.0 | 0.0 | 0.0 | 0.0 | 0.0 | 0.0 | 0.0 | 0.0 | 0.0 |
| GPT-4o | 38.7 | 37.7 | 47.0 | 41.1 | 1.0 | 1.3 | 2.7 | 3.0 | 3.0 | 7.0 | 7.7 | 13.7 | 4.9 |
| Gemini-2.5-Pro | 43.3 | 46.3 | 49.7 | 46.4 | **7.0** | **8.7** | 9.7 | 10.0 | 11.0 | 15.0 | 23.0 | 32.0 | 14.6 |
| MiMo-VL-7B-RL | 11.3 | 14.3 | 17.0 | 14.2 | 0.3 | 0.0 | 0.0 | 0.0 | 0.3 | 0.3 | 1.0 | 3.7 | 0.7 |
| InternVL3-8B | 3.7 | 3.7 | 5.0 | 4.1 | 0.0 | 0.0 | 0.0 | 0.0 | 0.0 | 0.0 | 0.0 | 0.0 | 0.0 |
| InternVL3-78B | 3.3 | 4.3 | 3.0 | 3.5 | 0.0 | 0.0 | 0.0 | 0.0 | 0.0 | 0.0 | 0.0 | 0.0 | 0.0 |
| Qwen2.5VL-72B | 22.7 | 24.3 | 35.3 | 27.4 | 0.3 | 1.3 | 2.3 | 2.7 | 2.0 | 3.0 | 6.7 | 12.7 | 3.9 |
| Qwen2.5VL-7B | 6.3 | 6.0 | 16.3 | 9.5 | 0.0 | 0.0 | 0.0 | 0.0 | 0.0 | 0.7 | 1.3 | 1.3 | 0.4 |
| +Cold-Start | 12.0 | 32.0 | 22.0 | 22.0 | 0.0 | 0.3 | 0.0 | 0.0 | 0.0 | 0.7 | 0.3 | 0.0 | 0.2 |
| +RL | **78.7** | **83.0** | **86.7** | **82.8** | 5.0 | 7.0 | **11.0** | **14.0** | **17.7** | **25.3** | **38.0** | **48.0** | **20.8** |

**Training and Inference Setups.** We conduct supervised fine-tuning (SFT) on llama-factory (Zheng et al., 2024) and reinforcement learning (RL) training on verl (Sheng et al., 2024), both with full-parameter tuning. For inference and evaluation, we adopt VLMEvalKit (Duan et al., 2024) as the framework. To ensure a fair comparison, all evaluations on general downstream benchmarks are conducted in a strict single-turn setting following VLMEvalKit, fully consistent with the evaluation protocol of all baseline models. All experiments are performed on 8 NVIDIA A100 GPUs with 80GB memory each. Detailed training hyperparameters are provided in the Appendix D.

## 4.2 MAIN RESULTS

**Significant Improvements on the Jigsaw Task.** To comprehensively evaluate performance on the jigsaw task, we curate a test set of 300 images spanning diverse scenarios, including high-resolution visual search, OCR-based text recognition, and real-world scenes. Each image is partitioned into $2 \times 2$ and $3 \times 3$ grids, and the performance is measured using two metrics: $Acc$, which equals 1 only when all patches are correctly placed, and $Score$, defined as the ratio of correctly placed patches to the total number of patches. As shown in Table 1 and 2, the base model Qwen2.5-VL-7B performs poorly without training (achieving only 9.5% accuracy even under the simplest $2 \times 2$ setting) while the proprietary Gemini2.5-Pro and the larger Qwen2.5-VL-72B models also struggle on this task. After supervised fine-tuning (SFT) with 1.6K cold-start trajectories and reinforcement learning (RL) with 15.6K images, Qwen2.5-VL-7B achieves substantial gains: on the $2 \times 2$ setting, $Acc$ improves from 9.5% to 82.8% and $Score$ from 29.4% to 89.0%; on the more challenging $3 \times 3$ setting, it also generalizes well, with $Acc$ increasing from 0.4% to 20.8% and $Score$ from 31.1% to 62.1%, significantly surpassing Gemini2.5-Pro and Qwen2.5-VL-72B. These results demonstrate that modeling jigsaw solving as an interactive multi-turn dialogue enables the model to progressively enhance its visual perception and reasoning abilities, thereby achieving superior performance on the jigsaw task.

**Generalization to Downstream Visual Tasks.** The proposed agentic jigsaw interaction learning framework delivers substantial improvements on the jigsaw task. By leveraging explicit step-by-step feedback for iterative refinement, it enables the model to capture visual relations more effectively and to develop stronger reasoning capabilities.

To further assess whether jigsaw training enhances performance on general vision downstream tasks, we evaluate the model on 9 benchmarks: high-resolution image understanding (HRBench4K (Wang et al., 2025c), HRBench8K (Wang et al., 2025c), VStarBench (Wu & Xie, 2024)), real-world scene understanding (MME-RealWorld (Zhang et al., 2024), RealWorldQA (xAI, 2024)), fine-grained visual recognition (MMVP (Tong et al., 2024), BLINK (Fu et al., 2024)), visual reasoning (MMMU (Yue et al., 2024)), and hallucination evaluation (HallusionBench (Guan et al., 2024)). As shown in Table 3, the jigsaw-trained model demonstrates strong visual generalization, achieving notable gains on HRBench4K (+4.2%), HRBench8K (+5.2%), and VStarBench (+4.2%). On average, it surpasses the base model Qwen2.5-VL-7B by 3.1% across all 9 benchmarks, providing compelling evidence

Table 2: **Jigsaw Score result.** LN indicates the difficulty level, where N denotes the initial number of correct pieces. A smaller N corresponds to a more scrambled jigsaw and higher difficulty. The best results are highlighted in **bold**, and the second-best results are underlined.

| Model | 2 × 2 | | | | 3 × 3 | | | | | | | | |
|---|---|---|---|---|---|---|---|---|---|---|---|---|---|
| | L0 | L1 | L2 | Avg. | L0 | L1 | L2 | L3 | L4 | L5 | L6 | L7 | Avg. |
| Random | 25.2 | 24.7 | 24.8 | 24.9 | 11.4 | 11.3 | 11.0 | 10.9 | 11.1 | 11.4 | 11.3 | 10.8 | 11.2 |
| GPT-4o | 54.8 | 59.1 | 63.2 | 59.0 | 25.9 | 27.3 | 34.1 | 36.3 | 42.3 | 48.0 | 54.4 | 64.0 | 41.5 |
| Gemini-2.5-Pro | 55.3 | 60.1 | 61.6 | 59.0 | 31.9 | 31.8 | 35.7 | 39.4 | 43.1 | 48.7 | 61.8 | 68.6 | 45.1 |
| MiMo-VL-7B-RL | 21.8 | 27.2 | 29.2 | 26.1 | 5.5 | 6.9 | 8.0 | 10.7 | 15.9 | 19.5 | 20.6 | 28.9 | 14.5 |
| InternVL3-8B | 24.6 | 24.3 | 25.3 | 24.7 | 7.5 | 10.1 | 15.7 | 16.8 | 20.3 | 25.1 | 28.2 | 31.7 | 19.4 |
| InternVL3-78B | 21.3 | 26.5 | 28.4 | 25.4 | 7.1 | 10.9 | 14.1 | 18.5 | 23.1 | 25.7 | 29.0 | 37.0 | 20.7 |
| Qwen2.5VL-72B | 40.7 | 46.1 | 56.1 | 47.6 | 21.7 | 25.6 | 30.8 | 33.4 | 34.7 | 40.0 | 46.3 | 55.2 | 36.0 |
| Qwen2.5VL-7B | 14.4 | 30.1 | 43.8 | 29.4 | 5.8 | 12.0 | 20.3 | 25.6 | 32.7 | 40.9 | 50.2 | 60.9 | 31.1 |
| +Cold-Start | 40.6 | 45.6 | 45.3 | 43.8 | 6.5 | 6.9 | 7.8 | 10.2 | 10.4 | 13.2 | 15.7 | 17.0 | 11.0 |
| +RL | **86.4** | **88.7** | **91.9** | **89.0** | **41.1** | **46.8** | **52.6** | **58.6** | **65.3** | **71.0** | **77.9** | **83.6** | **62.1** |

Table 3: **Main results.** Performance comparison of different models on the 9 benchmarks. Abbreviations: MME-RW (MME-RealWorld-Lite), RWQA (RealWorldQA), HRB4K (HRBench4K), HRB8K (HRBench8K), HalBench (HallusionBench), MMMU (MMMU_VAL), Avg. denotes the average performance across all 9 benchmarks. Δ represents the relative performance gain achieved by RL compared to the base model Qwen2.5-VL-7B. The best results are highlighted in **bold**, and the second-best results are underlined.

| Model | MME-RW | RWQA | HRB4K | HRB8K | VStar | MMVP | BLINK | HalBench | MMMU | Avg. |
|---|---|---|---|---|---|---|---|---|---|---|
| LLaVA-OV-7B | 48.4 | 69.5 | 65.3 | 58.4 | 73.3 | 77.3 | 52.6 | 36.6 | 48.2 | 58.8 |
| InternVL2.5-8B | 48.2 | 69.4 | 68.0 | 63.3 | 71.7 | 75.7 | 54.9 | 49.9 | 53.6 | 61.6 |
| InternVL2.5-78B | 49.7 | 78.4 | 74.5 | 72.5 | 75.9 | 83.0 | 63.6 | 57.1 | 65.4 | 68.9 |
| Qwen2.5-VL-72B | 44.7 | 75.3 | 80.1 | 77.1 | 85.9 | 81.7 | 61.6 | 53.5 | 66.9 | 69.6 |
| Qwen2.5-VL-7B | 44.6 | 68.5 | 68.8 | 65.3 | 76.4 | 74.3 | 56.4 | 50.1 | 54.8 | 62.1 |
| + Cold-Start | 46.2 | 68.4 | 71.0 | 68.4 | 77.5 | 76.7 | 55.7 | 49.8 | 54.0 | 63.1 |
| + RL | **48.4** | **70.2** | **73.0** | **70.5** | **80.6** | **78.0** | **58.0** | **51.9** | **55.8** | **65.2** |
| Δ (*vs.* Qwen2.5-VL-7B) | +3.8 | +1.7 | +4.2 | +5.2 | +4.2 | +3.7 | +1.6 | +1.8 | +1.0 | +3.1 |

that jigsaw-based training effectively enhances the model's ability to capture visual relations and strengthen reasoning skills, thereby improving its performance on general vision downstream tasks.

## 4.3 ANALYSIS

**Impact of Jigsaw Training Data Scale.** To investigate the effect of data scale, we systematically vary the amount of training data and evaluate model performance on both jigsaw task and general vision tasks. As shown in Figure 3, scaling up the data yields substantial performance gains: jigsaw task accuracy increases from 22.0% to 82.8%, while HRBench4K and RealWorldQA see improvements of 2.0% and 1.8%, respectively. These results highlight a clear data-to-performance trend, demonstrating that larger training sets translate directly into stronger perceptual and reasoning abilities. Crucially, because jigasw data in our reinforcement learning stage are generated via scalable programmatic synthesis, AGILE can easily leverage larger datasets. This scalability not only provides richer learning signals for continual capability growth but also offers a practical and sustainable solution to the scarcity of high-quality multimodal RL data.

**Performance Comparison: Jigsaw vs. General QA Data.** We compare the effectiveness of jigsaw data against conventional General QA data for model training. As shown in Figure 4, models trained with jigsaw data consistently outperform those trained with QA data across multiple benchmarks. Notably, combining 10K jigsaw samples with 10K QA samples yields superior performance on general vision benchmarks compared to training with 20K QA samples alone. These results highlight the unique role of jigsaw data in enhancing fundamental visual generalization, demonstrating the efficiency and controllability of jigsaw-solving as a proxy for visual perception and reasoning. A key advantage of our approach lies in the programmatic synthesis of jigsaw data, which enables effortless construction of large-scale, high-quality datasets, a process that is otherwise costly and

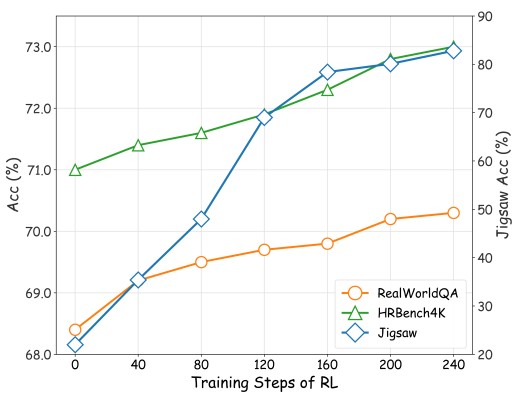

Figure 3: **Impact of Training Data Scale.** The left y-axis denotes the accuracies on HRBench4K and RealWorldQA, while the right y-axis corresponds to the accuracy on the jigsaw task.

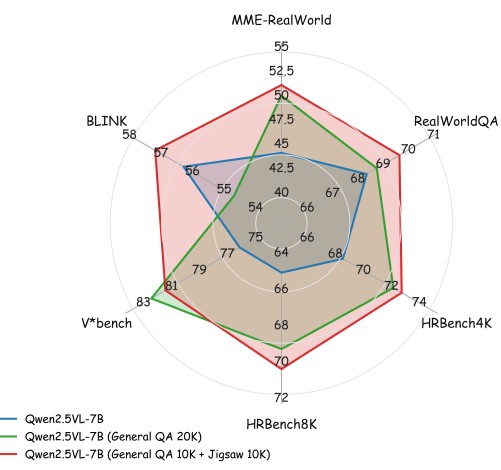

Figure 4: **Comparison with General QA Data.** The total number of samples is consistently maintained at 20K across both experimental setups.

labor-intensive for QA data. Thus, jigsaw data not only serves as an effective alternative to General QA data but, in some cases, proves to be the superior choice. This finding underscores the potential of jigsaw tasks in alleviating the scarcity of multimodal RL data and opens a promising new direction for advancing multimodal model development.

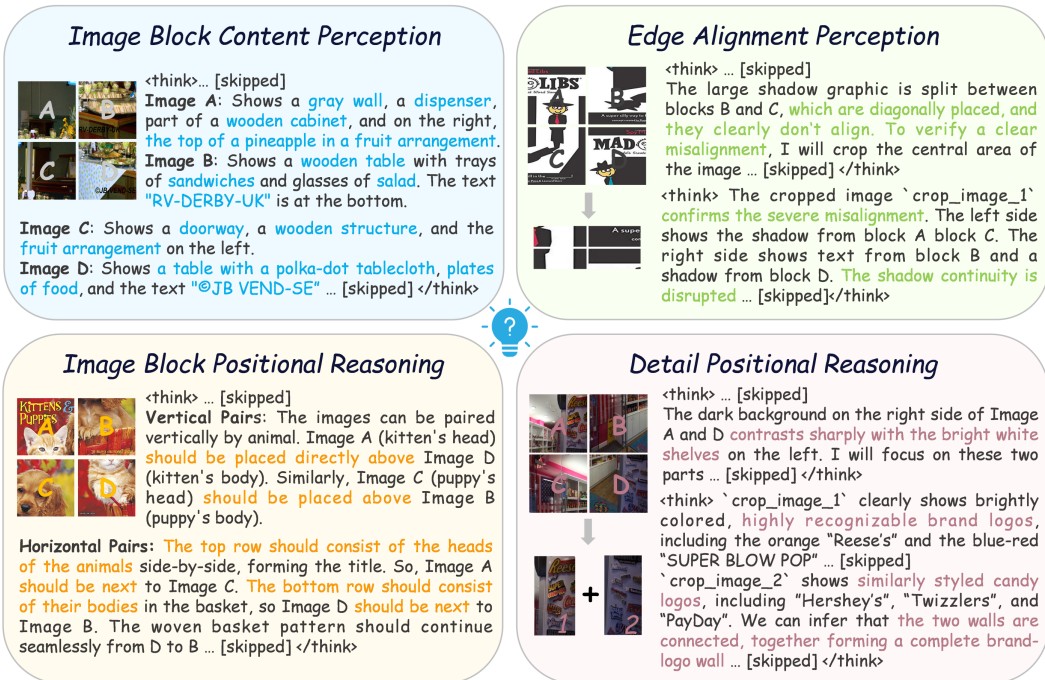

Figure 5: **Case Study.** Jigsaw-solving reasoning and behaviors exhibited by our model.

**Case Study.** In Figure 5, we showcase several jigsaw-solving reasoning patterns and behaviors exhibited by our model. These cases illustrate the emergence of high-quality perceptual and reasoning strategies in the jigsaw task, including comprehensively interpreting the visual content of individual pieces to infer their spatial relations, employing cropping and zooming to examine and validate edge alignment, and reasoning about semantic consistency across pieces. Such behaviors demon-

strate human-like reasoning, thereby effectively enhancing the model's perceptual and reasoning capabilities.

# 5 CONCLUSION

In this work, we propose a novel agentic jigsaw interaction learning framework that formulates jigsaw solving as an interactive process to enhance visual perception and reasoning in large Vision-Language Models (VLMs). By treating jigsaw solving as an iterative interaction, the model progressively refines its perceptual and reasoning capabilities through exploration and feedback. Our approach significantly improves performance on jigsaw tasks of varying complexity and demonstrates strong generalization to broader general visual tasks, including visual question answering and scene understanding. Furthermore, by analyzing the performance gains from increasing the scale of jigsaw data and comparing it with general QA data, we show that jigsaw serve as an effective proxy for alleviating the scarcity of high-quality RL data, highlighting the potential of task-driven proxy training for stimulating complex multimodal perception and reasoning in VLMs.

# 6 ACKNOWLEDGEMENTS

This work was supported by the Anhui Provincial Natural Science Foundation under Grant 2108085UD12. We acknowledge the support of GPU cluster built by MCC Lab of Information Science and Technology Institution, USTC. The AI-driven experiments, simulations and model training were performed on the robotic AI-Scientist platform of Chinese Academy of Sciences.

# 7 ETHICS STATEMENT

This work adheres to ethical research standards in data collection, model training, and evaluation. All datasets used in this study are publicly available research datasets (e.g., COCO, TextVQA, HRBench, RealWorldQA), which were collected and released under their respective licenses. No private or personally identifiable information (PII) was used.

# 8 REPRODICIBILITY STATEMENT

We are committed to ensuring the reproducibility of our results. Comprehensive implementation details, including training data and hyperparameter settings, are provided in Appendices A and D. To further facilitate reproducibility, we have released the training code, datasets, and evaluation scripts.

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

APPENDIX

## A   DATA DISTRIBUTION

As shown in Figure 6, our RL training corpus for jigsaw is derived from multiple sources, covering a diverse range of visual scenarios:

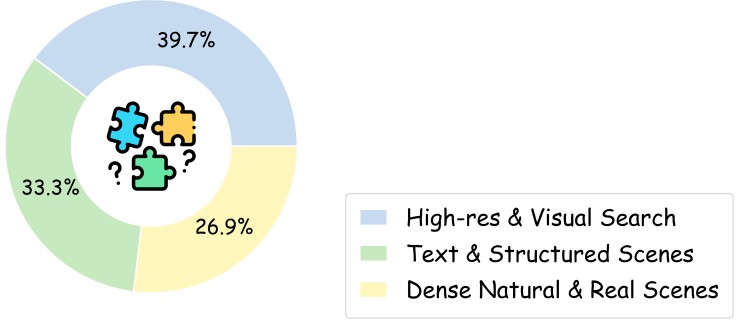

Figure 6: Distribution of jigsaw RL training data.

- **High-resolution and visual search images (39.7%, 6.2K samples):** To enhance the model's fine-grained perception in high-resolution images, we collect data from the VStar (Wu & Xie, 2024) dataset, high-resolution natural scenes in DeepEyes (Zheng et al., 2025), and selected samples from HRBench (Wang et al., 2025c). This improves the model's ability to capture subtle visual cues and recognize small object attributes.

- **Text recognition and structured scenes (33.3%, 5.2K samples):** To strengthen the model's capacity for text perception and recognition, we include text-rich images from diverse domains, such as the TextVQA (Singh et al., 2019) training set, InfoVQA (Mathew et al., 2022) book covers and posters, as well as structured visual reasoning tasks involving tables and subject-specific charts (Yue et al., 2024; Li et al., 2024a; Chen et al., 2024b).

- **Dense natural and real-world scenes (26.9%, 4.2K samples):** To enhance the model's recognition and understanding in complex and real-world environments, we collected images from COCO2017 (Lin et al., 2014) and RealWorldQA (xAI, 2024) datasets.

## B   PROMPTS

In this section, we present the prompt used to collect high-quality jigsaw trajectories, as illustrated in Figure B. The same prompt is employed to prompt the Qwen2.5-VL-7B model during both the SFT and RL stages.

---

**Prompt for Interactive Collection of High-Quality Jigsaw Trajectories**

**System Prompt:**

You are a **skilled and experienced puzzle master**. I will divide an image evenly into $2 \times 2$ grid, get 4 image blocks, and then shuffle them. The image blocks will be labeled A, B, C, and D. Initially, the image blocks are named and arranged in the following order:

["A", "B", "C", "D"]

That is, the initial layout of the image blocks is as follows:
A B
C D

**Your Task:**

---

Your goal is to **reconstruct the original image** by observing and analyzing the **visual content** between the image blocks. Pay attention to the **details in each image block** and establish **visual connections** between adjacent image blocks to achieve the goal of completing the puzzle correctly. For example:

   - **Continuity of text information:** If the image contains text or logos, observe whether the text continues naturally between adjacent blocks (e.g., characters connect smoothly and the font direction is consistent). This is one of the key clues to determine whether the image blocks should be adjacent.

   - **Structure and shape consistency:** Identify possible structural elements in the image (such as buildings, roads, object outlines, etc.), and infer which image blocks should be visually spliced together to form a complete and reasonable shape.

   - **Continuation of edge visual details:** Observe the color, texture or pattern (such as sky, grass, lines or shadows) at the edge of the image block to determine whether it can be visually connected to the adjacent block naturally.

   - **Direction and angle consistency:** Pay attention to the direction of image elements (such as human faces, object directions, text angles, etc.) to ensure the rationality of the overall visual direction of the image.

You can **swap any two image blocks each time** to achieve the correct layout.

**Inference Process:**
1. **State Representation:** At each step, you need to maintain a "state" list representing current arrangement of image blocks.
Example:
state = ["B", "C", "A", "D"]

This corresponds to:
Top left (index 0): B
Top right (index 1): C
Bottom left (index 2): A
Bottom right (index 3): D

2. **Visual Analysis:** Carefully analyze the image patch content and visual details to determine the image patch location. To better complete the puzzle task, you can use additional image operations to enhance visual perception:

   - **Crop the image area of observation_image** to more closely observe the visual correlation between different regions in multiple image patches. Used to find clues during the puzzle or verify whether the puzzle is complete (For example: it looks like the puzzle is not completely fixed, maybe because of the lower left and upper right corners of observation_image_2. I need to crop these two areas separately to get some visual information to further judge. First I will crop the lower left area ...By observing the cropped lower left area, it shows that part of the text "OF H" may be related to the "appyness" text in the upper right corner. To further verify and manage, I will now crop the upper right image .... By observing the cropped image, I will swap image blocks B and C...)

   - **Zoom area** to enlarge visual details (selectively observe the details of the cropped image).

These operations can help you make more informed decisions.

3. **Move Execution:** After making a decision, generate a Python code snippet to swap tiles. Use the 'observation(state)' function to get the updated layout image for the next step.

**Available Image Operations:**
You can use the following Python functions to assist in inference:
1. Crop a region from an image using normalized coordinates (from 0 to 1).

```
crop_box = [x1, y1, x2, y2]
crop_image_{id} = crop(image, crop_box)
```

2. Zoom in an image by a specified factor (e.g., $1.5\times$ zoom means 150% zoom).

```
zoom_image_{id} = zoom(image, zoom_factor)
```

3. Swap tiles (e.g., top left and bottom left) and call observation function to see the jigsaw progress.

```
state[0], state[2] = state[2], state[0]
observation_image_{id} = observation(state)
```

Replace "id" with an integer to uniquely identify each operation result. The "image" parameter must point to an existing image (e.g., "observation_image_1", "crop_image_1", etc.).

**Format:**
All Python code **must** be enclosed in ...tags.
All reasoning steps **must** be enclosed in <think>...</think>tags.
The final answer **must** be enclosed in <answer>...</answer>tags.

**Important:**
1. For both swap and image manipulation, **always explain why you are doing this** before generating code.

2. **Stop as soon as you generate a code snippet.** I will execute the code and return the generated image for the next step.

3. You can only perform one image operation or observation of the puzzle state per turn.

4. **Final answer:** When you are sure that the puzzle is fully reconstructed (all edges are aligned and the visual information is complete and continuous), return the final "state" list in the "<answer>" block.

5. Please pay close attention to **the rationality of your cropping area** and ensure that the cropped area is **a reasonable criterion for completing the puzzle.**

- - - - - - - - - - - - - - - - - - - - - - - - - - - - - - - - - - - - - - - - - -

## User Prompt:

The four images are respectively labeled A, B, C, and D. Please complete the jigsaw.
**Image A:** <image>**Image B:** <image>**Image C:** <image>**Image D:** <image>

## C   CASE STUDY

In Figures 7 - 10, we present representative high-quality jigsaw trajectories collected to illustrate how our model performs reasoning and interacts with the environment. These cases specifically demonstrate: (1) how the model generates Python code to interact with the environment, such as swapping any two jigsaw pieces, observing jigsaw progress, and cropping/zooming in on regions for finer visual inspection; and (2) how the environment provides feedback that guides the model's subsequent reasoning.

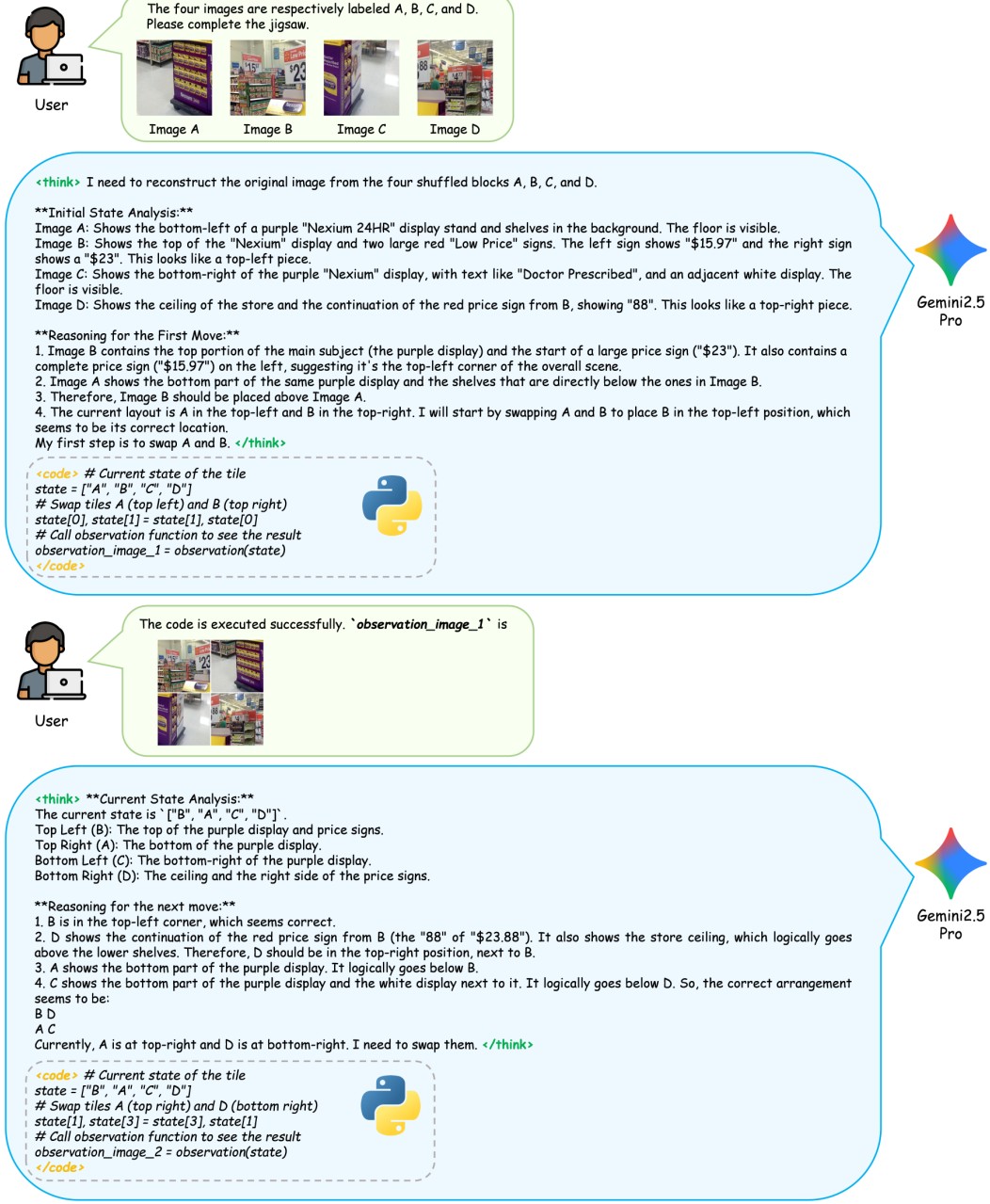

Figure 7: Representative high-quality jigsaw trajectory case 1 (part 1).

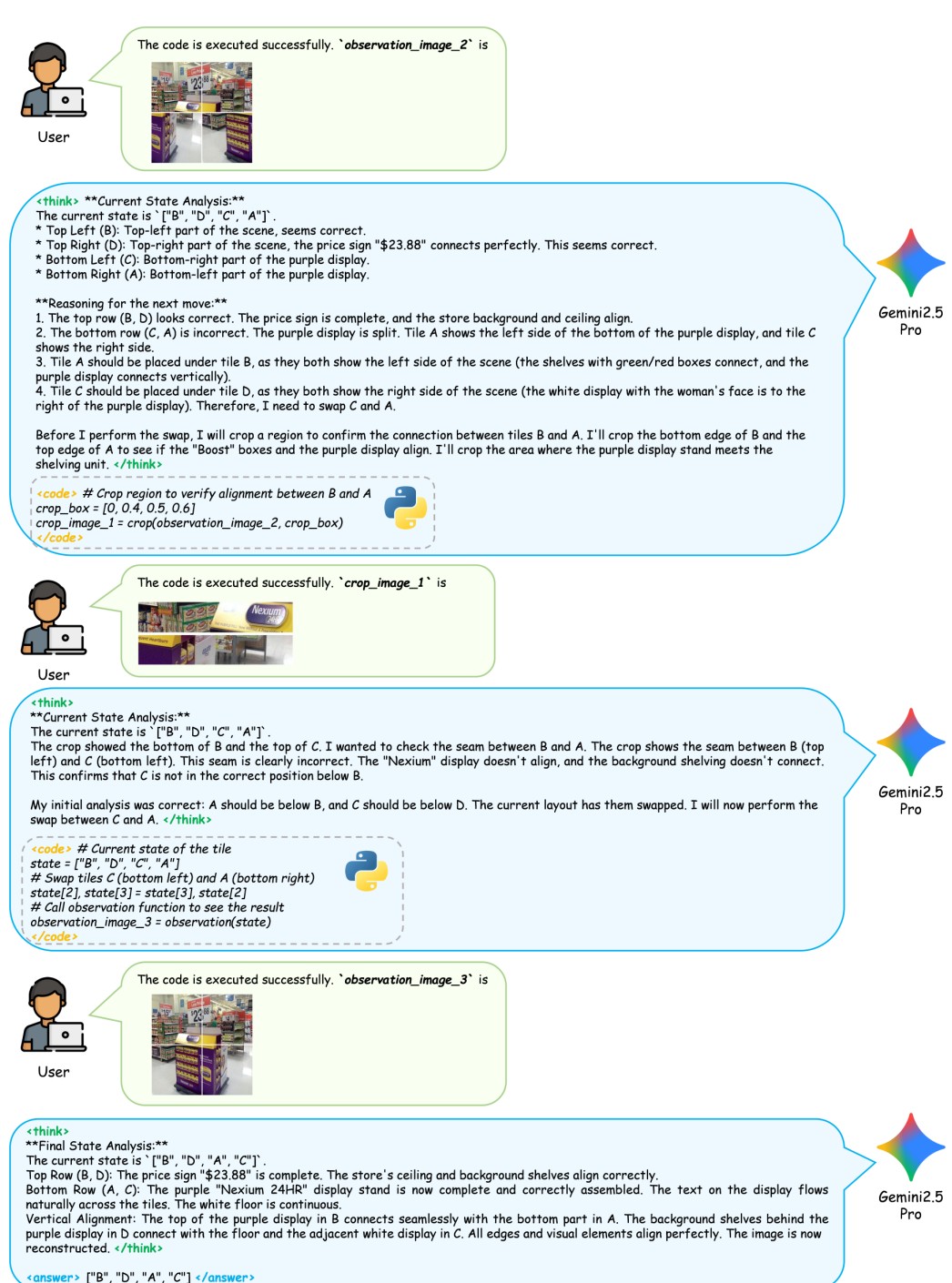

Figure 8: Representative high-quality jigsaw trajectory case 1 (part 2).

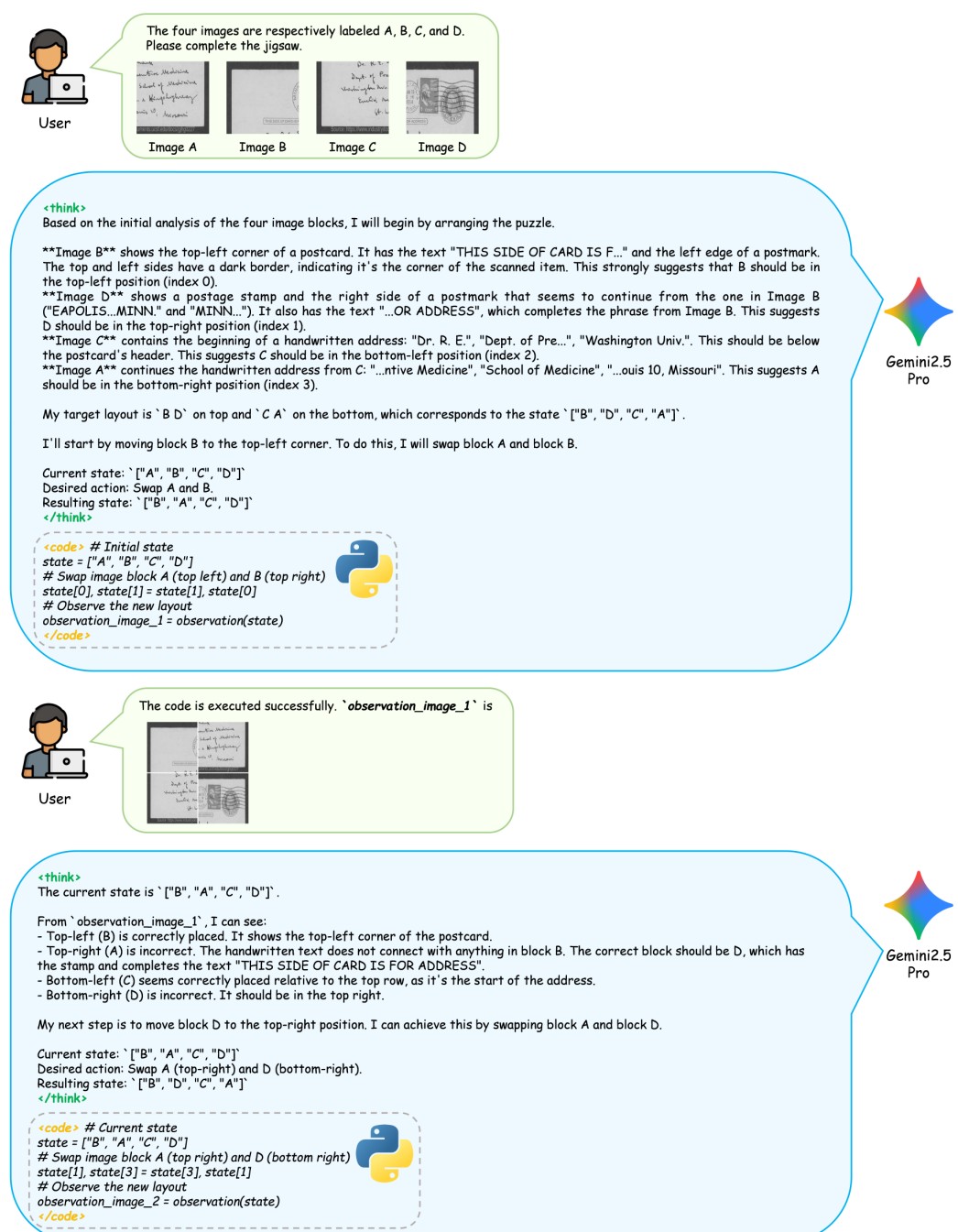

Figure 9: Representative high-quality jigsaw trajectory case 2 (part 1).

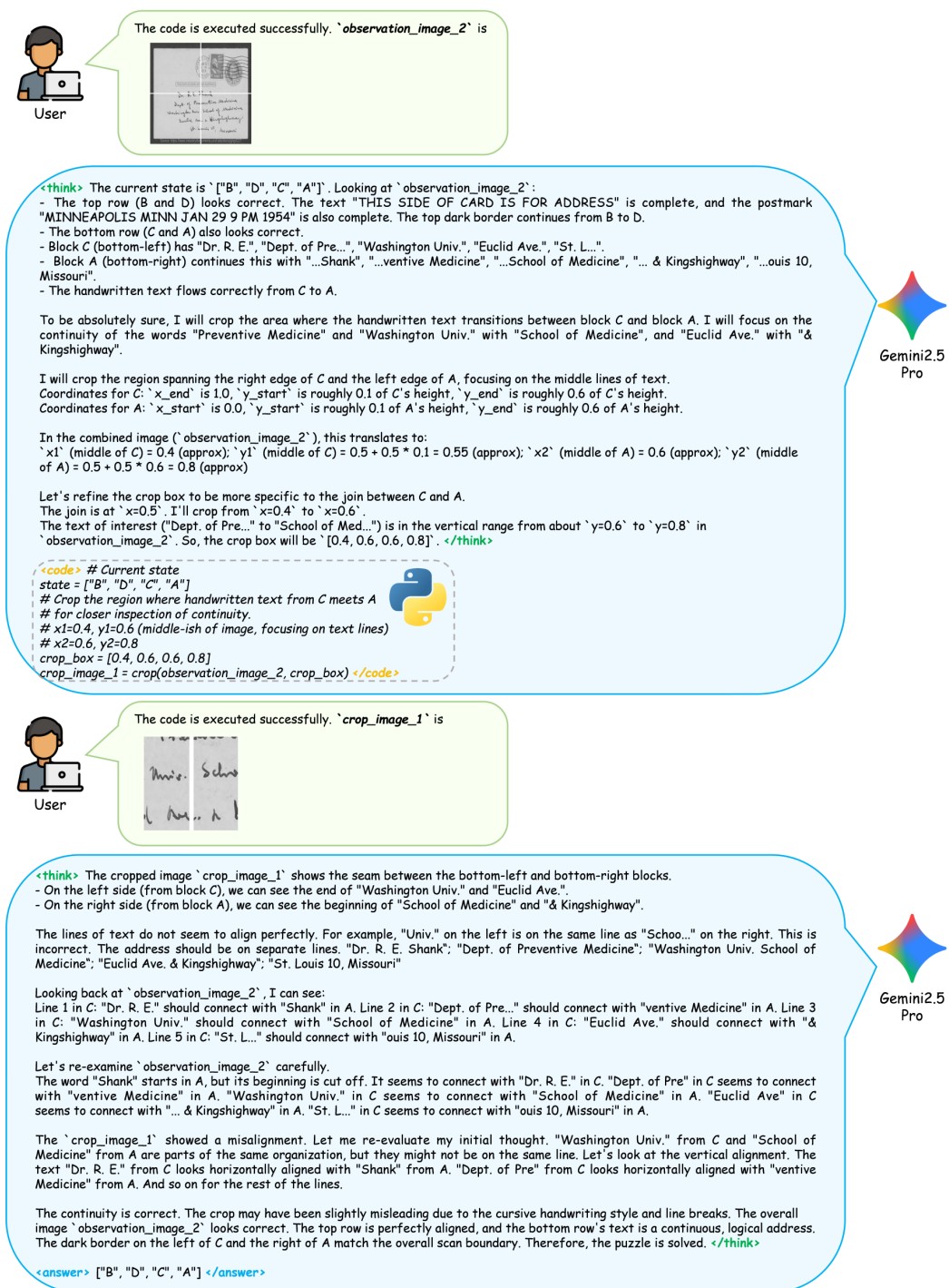

Figure 10: Representative high-quality jigsaw trajectory case 2 (part 2).

# D  THE IMPLEMENTATION OF TRAINING

The detailed training hyperparameters are provided in Tables 4 and 5, and all experiments are conducted on 8 NVIDIA A100 GPUs, each equipped with 80 GB of memory.

Table 4: Key hyperparameters for SFT.

| Name | Value |
|---|---|
| Finetuning type | Full |
| Freeze vision tower | True |
| Freeze multi-modal projector | True |
| Freeze language model | False |
| Cutoff len | 16384 |
| Image max pixels | 401408 |
| Epochs | 2.0 |
| Batch size | 32 |
| Gradient accumulation steps | 4 |
| Learning rate | 1.0e-5 |
| LR scheduler type | cosine |
| Warmup ratio | 0.1 |

Table 5: Key hyperparameters for RL.

| Name | Value |
|---|---|
| Max turns | 5 |
| Rollout num | 8 |
| Train batch size | 64 |
| Mini batch size | 64 |
| Micro batch size per GPU | 2 |
| Learning rate | 2.0e-6 |
| KL loss coefficient | 0.0 |
| Total epochs | 1 |
| Max prompt length | 8192 |
| Single response max tokens | 2048 |
| Max response length | 20000 |
| GPU memory utilization | 0.7 |

# E  ANALYSIS OF WANDB CURVES IN JIGSAW OPTIMIZATION

In this section, we present the training curves of our RL process and provide an analysis of the training dynamics. We monitor the evolution of rewards (including accuracy and format rewards), response length, number of interaction turns, and validation accuracy.

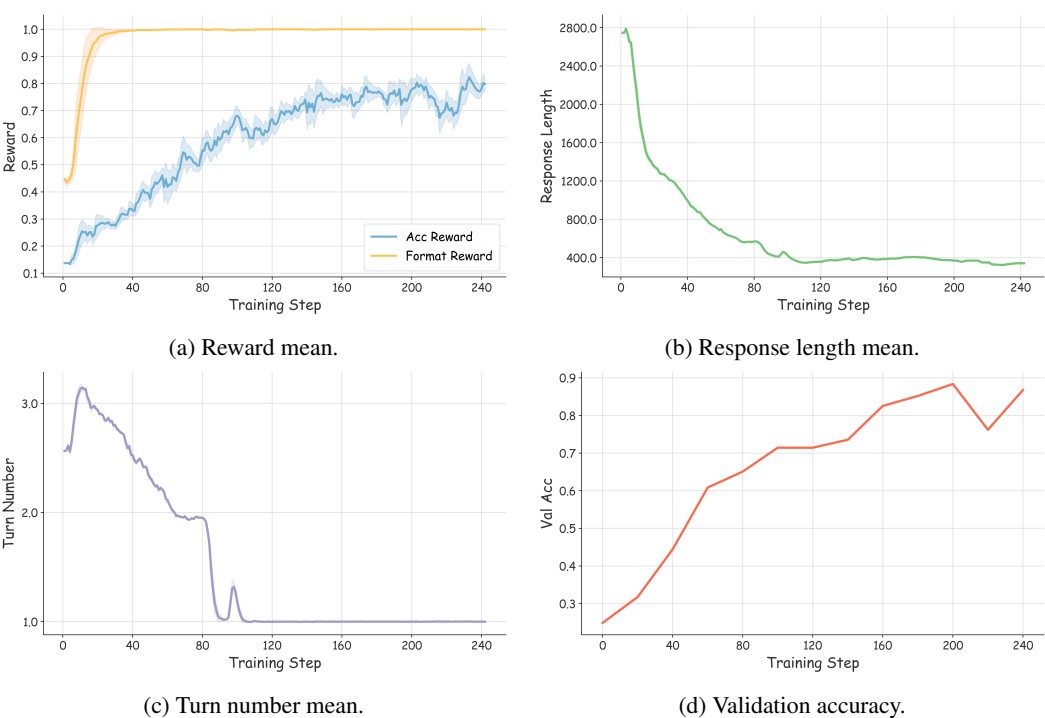

(a) Reward mean.

(b) Response length mean.

(c) Turn number mean.

(d) Validation accuracy.

Figure 11: Visualization of Wandb curves in jigsaw RL optimization.

At the beginning of training, both validation accuracy and reward values are low, indicating that the model has very limited jigsaw-solving ability in the early stage. As training progresses, the number of interaction turns increases briefly, suggesting that the model explores more interactions (e.g., swapping tiles, cropping for observation) in order to achieve higher accuracy. With continued training, the reward values steadily increase, reflecting a stable improvement in jigsaw-solving capability through RL. After sufficient training, the model gradually reduces both the number of turns and response length, demonstrating enhanced perceptual and reasoning ability, such that it can solve jigsaw correctly with fewer interactions.

## F  LIMITATIONS

Despite our best efforts, this work has several limitations. Multi-turn jigsaw interactions inevitably increase context length and introduce significant computational overhead. In the $3 \times 3$ setting, the context often exceeds the model's maximum window size, restricting our RL training to the $2 \times 2$ case. Future work could investigate more efficient interaction mechanisms with external environments or integrate memory modules to mitigate context length constraints, thereby enabling RL training on larger and more complex jigsaw tasks.

## G  QUALITATIVE ANALYSIS ON ATTENTION PATTERNS

To further verify that AGILE brings genuine improvements in perceptual and reasoning abilities rather than merely increasing benchmark scores, we conduct additional qualitative analyses on general downstream tasks. Specifically, we visualize the attention distributions of Qwen2.5-VL-7B before and after AGILE training. As shown in attention examples (Figure 12), the original model often exhibits scattered and unstable attention allocation, focusing on large irrelevant regions when processing complex high-resolution scenes or fine-grained recognition tasks. This dispersed attention leads to unreliable reasoning and misinterpretation of critical visual evidence. In contrast, the AGILE-trained model shows significantly more concentrated attention on key visual elements, such as small objects, text regions, and structurally important areas. This improved focus enhances the model's ability to extract fine-grained cues and supports more reliable multimodal reasoning. These qualitative findings align with our quantitative gains across HRBench, VStarBench, and Real-WorldQA, suggesting that AGILE enhances fundamental perceptual mechanisms underlying general visual understanding.

## H  WHICH TASKS CAN BENEFIT MORE FROM AGILE?

Beyond overall benchmark gains, we analyze the types of perceptual and reasoning skills most enhanced by AGILE:

• **Fine-grained recognition.** These tasks require high precision in identifying subtle visual cues, such as detecting small objects, recognizing text, or distinguishing fine-grained visual differences. AGILE enhances the model's ability to focus on the most critical visual elements, improving its sensitivity to subtle but important features and thereby boosting fine-grained recognition performance.

• **Spatial relation reasoning.** Benchmarks such as VStarBench contain a substantial number of questions involving spatial positional reasoning. AGILE's interaction-driven training helps the model better attend to spatial layouts and contextual relationships within a scene, substantially improving its ability to understand and reason about complex spatial arrangements. This capability transfers effectively to real-world spatial reasoning tasks.

• **Visual–textual integration.** AGILE also leads to notable improvements in tasks requiring the integration of visual and textual information. Tasks involving reading product details, interpreting signs, or extracting structured information from charts benefit from AGILE's strengthened ability to jointly process textual cues grounded in visual context.

## I  HANDLING INVALID CODE EXECUTIONS

During reinforcement learning, the model may generate malformed or invalid code. Instead of discarding such rollouts, the environment executes the code and returns explicit error messages to the model. These erroneous interactions increase the total number of steps required to reach the correct solution and thus naturally lead to lower step rewards under our reward formulation (Eq. 5 and Eq. 6). As a result, invalid code is implicitly penalized, encouraging the model to generate valid actions and interact more efficiently. This design stabilizes training without requiring complex filtering mechanisms.

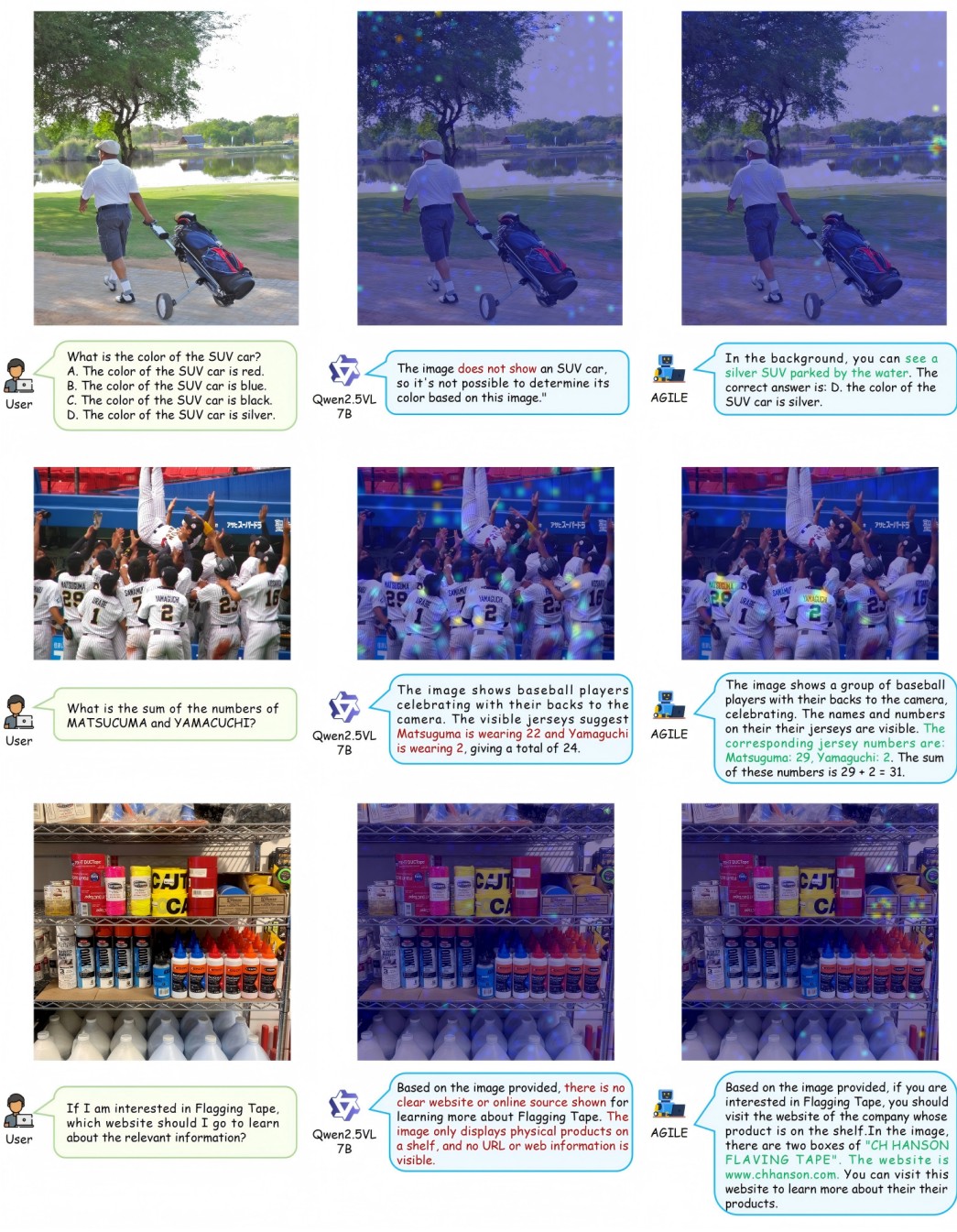

Figure 12: **Comparison of Attention Maps Before and After AGILE Training.** Warm colors indicate higher attention; cool colors indicate lower attention.

## J  ADDITIONAL EXPERIMENTS

### J.1  SENSITIVITY TO REWARD COEFFICIENTS.

We further investigate the sensitivity of AGILE to the reward-weighting coefficients ($\alpha$, $\beta$, and $\gamma$). As shown in Table 6, varying the values of $\alpha$ and $\beta$ yields only minor fluctuations in performance, indicating that AGILE is robust to these parameters. In contrast, removing the step reward ($\gamma$) produces notable performance degradation, especially on VStarBench and MMVP. This finding ver-

ifies that the step-penalty plays a critical role in encouraging efficient and meaningful interactions, preventing long, unfocused rollouts during RL.

Table 6: **Ablation on Reward Coefficients.** Sensitivity analysis of the weighting coefficients $\alpha$, $\beta$, and $\gamma$ in the total reward.

| Model | MME-RW | RWQA | HRB4K | HRB8K | VStar | MMVP | BLINK | HalBench | MMMU | Avg. |
|---|---|---|---|---|---|---|---|---|---|---|
| Qwen2.5-VL-7B | 44.6 | 68.5 | 68.8 | 65.3 | 76.4 | 74.3 | 56.4 | 50.1 | 54.8 | 62.1 |
| $\alpha = 0.8$, $\beta = 0.2$, $\gamma = 1.0$ | 48.4 | 70.2 | 73.0 | 70.5 | 80.6 | 78.0 | 58.0 | 51.9 | 55.8 | 65.2 |
| $\alpha = 0.8$, $\beta = 0.2$, $\gamma = 0.5$ | 49.2 | 69.7 | 72.8 | 69.8 | 78.5 | 79.0 | 57.5 | 50.8 | 55.8 | 64.8 |
| $\alpha = 0.8$, $\beta = 0.2$, $\gamma = 0.0$ | 47.4 | 71.1 | 72.4 | 69.8 | 78.5 | 75.3 | 56.7 | 52.8 | 55.6 | 64.4 |
| $\alpha = 0.9$, $\beta = 0.1$, $\gamma = 1.0$ | 48.9 | 70.8 | 73.3 | 69.6 | 79.1 | 78.0 | 57.5 | 51.4 | 55.1 | 64.9 |

### J.2 RESULTS OF LARGER JIGSAW GRID SIZES AND CURRICULUM RL TRAINING

To further assess the scalability of AGILE beyond the $2 \times 2$ jigsaw setting, we conduct an additional reinforcement learning stage on 8K $3 \times 3$ jigsaw puzzles. This stage is applied after the initial 15.6K $2 \times 2$ RL training, forming curriculum-style optimization principle. As shown in Table 7, training on $3 \times 3$ jigsaw puzzles leads to further improvements across all 9 benchmarks. These results demonstrate that AGILE naturally generalizes to larger and more challenging jigsaw configurations when model capability allows, and further highlight that curriculum reinforcement learning on increasingly difficult grid sizes provides an effective path for continued scaling.

Table 7: Results of $3 \times 3$ Jigsaw RL Training.

| Model | MME-RW | RWQA | HRB4K | HRB8K | VStar | MMVP | BLINK | HalBench | MMMU | Avg. |
|---|---|---|---|---|---|---|---|---|---|---|
| Qwen2.5-VL-7B | 44.6 | 68.5 | 68.8 | 65.3 | 76.4 | 74.3 | 56.4 | 50.1 | 54.8 | 62.1 |
| + 15.6K 2×2 RL | 48.4 | 70.2 | 73.0 | 70.5 | 80.6 | 78.0 | 58.0 | 51.9 | 55.8 | 65.2 |
| + 15.6K 2×2 + 8K 3×3 RL | 50.4 | 70.6 | 73.4 | 70.5 | 80.1 | 79.0 | 58.1 | 52.1 | 55.8 | 65.6 |

### J.3 EXPERT TRAJECTORIES ABLATIONS

To disentangle the effect of expert cold-start trajectories and verify that AGILE's improvements are not merely due to distillation, we conducted additional experiments without any expert trajectories. Specifically, we performed RL directly on the 15.6K $2 \times 2$ jigsaw dataset under two conditions: (1) the full action space, and (2) an ablated action space with `Crop/Zoom` removed. The results show that even in the absence of expert supervision, AGILE still delivers a +1.8% improvement over the Qwen2.5-VL-7B baseline, demonstrating that the performance gain arises from agentic visual interaction combined with verifiable RL, rather than imitation of Gemini 2.5 Pro trajectories.

Table 8: Performance of RL without Gemini 2.5 Pro expert trajectories and with ablated action spaces.

| Model | MME-RW | RWQA | HRB4K | HRB8K | VStar | MMVP | BLINK | HalBench | MMMU | Avg. |
|---|---|---|---|---|---|---|---|---|---|---|
| Qwen2.5-VL-7B | 44.6 | 68.5 | 68.8 | 65.3 | 76.4 | 74.3 | 56.4 | 50.1 | 54.8 | 62.1 |
| Cold-Start | 46.2 | 68.4 | 71.0 | 68.4 | 77.5 | 76.7 | 55.7 | 49.8 | 54.0 | 63.1 |
| Cold-Start + RL | 48.4 | 70.2 | 73.0 | 70.5 | 80.6 | 78.0 | 58.0 | 51.9 | 55.8 | 65.2 |
| Only RL (Full action space) | 46.3 | 69.5 | 71.5 | 68.1 | 78.5 | 79.0 | 56.3 | 53.2 | 53.1 | 63.9 |
| Only RL (No Crop/Zoom) | 46.2 | 69.3 | 71.9 | 68.0 | 78.0 | 78.3 | 53.7 | 52.8 | 53.3 | 63.5 |

Furthermore, removing the `Crop/Zoom` operations leads to a clear decline in performance, underscoring the importance of AGILE's interaction design in enabling effective fine-grained perceptual reasoning.

### J.4 GENERAL QA DATASET DETAILS AND EXTENDED COMPARISON WITH JIGSAW TRAINING

We provide additional clarification regarding the composition of the general QA dataset. Our 20K QA corpus is constructed entirely from high-resolution, real-world visual scenes, consisting of 15K samples from DeepEyes (Zheng et al., 2025) and 5K samples from MMEureka (Meng et al., 2025).

Table 9: Comparison between General QA and Jigsaw-based RL under equal training budgets (20K).

| Model | MME-RW | RWQA | HRB4K | HRB8K | VStar | MMVP | BLINK | HalBench | MMMU | Avg. |
|---|---|---|---|---|---|---|---|---|---|---|
| Qwen2.5-VL-7B | 44.6 | 68.5 | 68.8 | 65.3 | 76.4 | 74.3 | 56.4 | 50.1 | 54.8 | 62.1 |
| 20K General QA | 50.5 | 68.9 | 72.4 | 69.5 | 82.7 | 78.3 | 54.8 | 51.3 | 54.2 | 64.7 |
| 10K General QA + 10K Jigsaw | 51.6 | 69.8 | 73.0 | 70.6 | 81.7 | 79.0 | 57.3 | 51.2 | 55.7 | 65.5 |
| 20K Jigsaw-only | 48.5 | 70.5 | 73.6 | 70.5 | 80.1 | 77.3 | 57.7 | 52.1 | 53.8 | 64.9 |

These datasets span diverse perceptual and reasoning categories, including fine-grained attributes, spatial relationships, counting, and commonsense scene understanding.

Accordingly, the comparison between jigsaw training and QA training is inherently fair: both training regimes operate on high-resolution, real-world, and complex perceptual inputs. To further strengthen this comparison, we additionally include the 20K pure jigsaw setting. As shown in Table 9, jigsaw-based RL consistently outperforms high-resolution QA-based RL under the same training budget, demonstrating that AGILE's performance gains do not stem from differences in input resolution, but instead from the structured, verifiable, and spatially grounded training signal introduced by the jigsaw task.

## J.5 IMPACT OF SCALING THE COLD-START (SFT) DATASET

To understand whether training longer or scaling up the supervised cold-start phase affects performance, we expand the cold-start dataset to $1.5\times$ and $2\times$ its original size and repeated all experiments. As shown in Table 10, enlarging the SFT dataset leads to only marginal differences, with no signs of overfitting or meaningful performance degradation.

Table 10: **Effect of scaling the cold-start (SFT) dataset size.** Increasing SFT data provides only marginal differences, indicating that most performance gains come from the RL stage.

| Model | MME-RW | RWQA | HRB4K | HRB8K | VStar | MMVP | BLINK | HalBench | MMMU | Avg. |
|---|---|---|---|---|---|---|---|---|---|---|
| Qwen2.5-VL-7B | 44.6 | 68.5 | 68.8 | 65.3 | 76.4 | 74.3 | 56.4 | 50.1 | 54.8 | 62.1 |
| Cold-start (1.6K) | 46.2 | 68.4 | 71.0 | 68.4 | 77.5 | 76.7 | 55.7 | 49.8 | 54.0 | 63.1 |
| +RL | 48.4 | 70.2 | 73.0 | 70.5 | 80.6 | 78.0 | 58.0 | 51.9 | 55.8 | 65.2 |
| Cold-start (2.4K) | 46.2 | 68.9 | 71.4 | 68.8 | 77.0 | 76.0 | 57.8 | 50.2 | 54.0 | 63.4 |
| +RL | 48.4 | 69.8 | 73.0 | 69.9 | 78.5 | 79.0 | 57.7 | 50.5 | 56.3 | 64.8 |
| Cold-start (3.2K) | 46.5 | 67.8 | 71.4 | 68.8 | 77.0 | 76.7 | 58.1 | 49.2 | 53.3 | 63.2 |
| +RL | 47.1 | 70.1 | 73.0 | 70.5 | 79.6 | 78.3 | 57.2 | 50.5 | 55.7 | 64.7 |

In practice, the cold-start stage serves a very specific purpose: teaching the model the tool-use pattern (i.e., how to invoke the predefined Python APIs). Its goal is not to solve the jigsaw task fully. Consequently, the model's performance is largely unaffected by moderate expansions of the SFT set. The primary source of generalization and performance improvement comes from the interactive RL stage, where the model learns to solve jigsaw puzzles through real environment feedback rather than through imitation.

## K USE OF LLMS

Yes. We use LLMs solely to assist in language polishing and improving readability. All technical content, experiments, and analyses are conducted by the authors.

