# OpenReview forum: "Agentic Jigsaw Interaction Learning for Enhancing Visual Perception and Reasoning in Vision-Language Models"
_ICLR.cc/2026/Conference — ICLR 2026 Poster_

### Official Review · Reviewer_yrdc · 2025-10-31

**Soundness:** 3
**Presentation:** 2
**Contribution:** 2
**Rating:** 2
**Confidence:** 5

**Summary:**

This paper tackles the problem that current VLMs struggle with. For basic perceptual and reasoning tasks like 2x2 jigsaw puzzles, they perform randomly. To fix this, the authors created AGILE, a framework that turns jigsaw solving into an interactive process. The model generates code to do actions like swapping pieces, cropping, or zooming in, and gets real-time visual feedback from the environment to keep improving. The model is trained with high-quality cold-start trajectories and jigsaw images, and the results are pretty impressive: the model’s accuracy on 2x2 jigsaws jumped from 9.5% to 82.8%, and it also got 3.1% better on average across 9 general vision tasks, showing it can generalize well beyond just solving puzzles.

**Strengths:**

1. The authors use a clever way to solve the data scarcity issue for multimodal reinforcement learning: code-based data generation for scalable data curation.

2. Interactive training design is impressive: by making the model actively engage with the jigsaw environment step by step, it builds real perceptual and reasoning skills instead of just memorizing patterns.

3. General improvement on different visual tasks validates the effectiveness and generalization of the proposed method.

**Weaknesses:**

1. The author did not mentioned comparisons with direct apply GRPO or other RL training algorithms. As RL can improve the performance of VLMs, I would like to know the real contribution of jigsaw training. Current improvement is combined with RL training.

2. The authors should provide more details about their experiments regarding the comparsion with general QAs. What does general QA dataset consist of? Also, I would like to know the result if I use high resolution general QA dataset to train the model. How it will perform? This will help others validate the contribution of using jigsaw for training more clear.

3. I think the auhors should provide more on the jigsaw bench. What if giving each patch a unique ID and let the model to sort the patch to the correct order? Will it improve the overall performance. I wonder if the code excecution is the best way of doing this? How about using ID and sort them, and then use this to train the model. VLMs are not good at coding, which will make them harder to solve problems in such settings. More details and comparisons are needed.

4. Cold-Start only use 1.6K data and RL using 16K data, what if training the model longer during SFT? will the improvements become fewer? Comparisons are needed regarding the scaling of training data.

**Questions:**

As stated in weakness part.

---

> ### Author Response · Authors · 2025-11-19
> **Detailed Rebuttal (1/3)**
>
> We thank the reviewer for acknowledging our contributions and for offering helpful feedback. We have conducted additional experiments and revised the paper accordingly. In the following rebuttal, we try to address each of your concerns to avoid potential misunderstandings.
>
>
> > **W1:** The author did not mentioned comparisons with direct apply GRPO or other RL training algorithms. As RL can improve the performance of VLMs, I would like to know the real contribution of jigsaw training. Current improvement is combined with RL training.
>
> We sincerely thank the reviewer for raising this important concern. We would like to clarify that our paper **already evaluates the contribution of jigsaw training independent of the RL algorithm**. As described in **Sec. 4.3 "Performance Comparison: Jigsaw vs. General QA Data" (Line 430)**, we directly compare the following two GRPO training settings under the same training budget (20K):
>
> - **20K General QA (GRPO)**
> - **10K General QA + 10K Jigsaw (GRPO)**
>
> This comparison isolates the effect of replacing general QA data with jigsaw data while keeping the RL algorithm, model architecture, and training steps identical. The results show that **jigsaw-based RL consistently outperforms general QA RL**, demonstrating that the improvement comes from the training signal rather than RL itself.
>
> To further address the reviewer's concern, we additionally conducted a new experiment using:
>
> - **20K pure jigsaw data (GRPO)**
>
> The new results confirm our conclusion: **both the equal-size replacement setting (10K Jigsaw) and the 20K Jigsaw-only setting outperform 20K General QA**, clearly illustrating the intrinsic value of jigsaw training as a more informative and structured visual-spatial reasoning signal.
>
> |                                    | MME-RW | RWQA | HRB4K | HRB8K | VStar | MMVP | BLINK | HalBench | MMMU | Avg. |
> | :--------------------------------: | :----: | :--: | :---: | :---: | :---: | :--: | :---: | :------: | :--: | :--: |
> |           Qwen2.5-VL-7B            |  44.6  | 68.5 | 68.8  | 65.3  | 76.4  | 74.3 | 56.4  |   50.1   | 54.8 | 62.1 |
> |       20K General QA (GRPO)        |  50.5  | 68.9 | 72.4  | 69.5  | 82.7  | 78.3 | 54.8  |   51.3   | 54.2 | 64.7 |
> | 10K General QA + 10K Jigsaw (GRPO) |  51.6  | 69.8 | 73.0  | 70.6  | 81.7  | 79.0 | 57.3  |   51.2   | 55.7 | 65.5 |
> |    20K pure jigsaw data (GRPO)     |  48.5  | 70.5 | 73.6  | 70.5  | 80.1  | 77.3 | 57.7  |   52.1   | 53.8 | 64.9 |
>
> We have included this new experiment and discussion in **Table 9** (**Appendix I.4**) to make the contribution of jigsaw training even clearer.
>
> ******************
>
> > **W2:** The authors should provide more details about their experiments regarding the comparsion with general QAs. What does general QA dataset consist of? Also, I would like to know the result if I use high resolution general QA dataset to train the model. How it will perform? This will help others validate the contribution of using jigsaw for training more clear.
>
> We thank the reviewer for the constructive questions regarding the composition of the general QA dataset and the role of high-resolution QA data.
>
> In our work, the **20K General QA dataset** is collected from two recent, high-quality, high-resolution visual QA sources:
>
> - 15K high-resolution scene QA samples from **DeepEyes** [1],
> - 5K high-resolution perception and reasoning QA samples from **MM-EUREKA** [2].
>
> Thus, the "general QA" we refer to is already built on high-resolution real-world scenes, covering object attributes, spatial relations, counting, commonsense reasoning, and factual scene understanding.
>
> The comparison presented in Sec. 4.3 (as also discussed in our response to **W1**) constitutes a **direct evaluation between high-resolution QA-based RL training and high-resolution jigsaw-based RL training**, conducted under strictly matched training budgets. The results clearly highlight the advantages of AGILE training in this fair setting.
>
> We have incorporated explicit descriptions of the dataset sources in **Appendix I.4** and clarified that our general QA training set is already composed of high-resolution samples. This ensures that the jigsaw-versus-QA comparison is both fair and fully interpretable.
>
> ******************

---

> > ### Author Response · Authors · 2025-11-19
> > **Detailed Rebuttal (2/3)**
> >
> > > **W3:** I think the auhors should provide more on the jigsaw bench. What if giving each patch a unique ID and let the model to sort the patch to the correct order? Will it improve the overall performance. I wonder if the code excecution is the best way of doing this? How about using ID and sort them, and then use this to train the model. VLMs are not good at coding, which will make them harder to solve problems in such settings. More details and comparisons are needed.
> >
> > Thank you for the question. We believe your concern arises from a misunderstanding of our jigsaw benchmark and the role of code execution. **Section 3.1** of our manuscript provides a detailed description of the task design, including data generation and allowed agent-environment interactions. Additionally, **Appendix B and C** present the full prompts and jigsaw case examples. To totally address your concerns, we offer the following clarifications and analyses:
> >
> > 1. Our benchmark already includes the patch-ID sorting setting in your comment. In our evaluation, each patch is assigned a fixed-order ID (A-D for 2 $\times$ 2), and the model is required to output the correct ID sequence. As shown in Table 1, state-of-the-art VLMs perform poorly under this configuration (e.g., Qwen2.5-VL-7B: 9.5%, Qwen2.5-VL-72B: 27.4%). These results indicate that **single-pass ID sorting already exposes a fundamental limitation** of current VLMs: the lack of reliable tile-level perception and spatial reasoning.
> >
> > 2. Solving jigsaw puzzles requires comparing **local textures, boundaries, and semantic continuities** across tiles. It is extremely challenging to infer all local relationships implicitly within a single step, even for frontier VLMs. Without interactive operations such as o**bserve, crop/zoom, compare, or swap** the model cannot selectively focus on ambiguous tiles or iteratively verify predictions. This motivates our **agent-based training framework**, which is also a core contribution of AGILE. The multi-step structure transforms a globally challenging problem into a **locally verifiable interaction loop**, leading to substantially more stable and effective learning.
> >
> > 3. We agree that VLMs are not strong general-purpose coders. Therefore, in AGILE, code generation is restricted to a small predefined **Python action API** (e.g., swap, observe). The model does not need to produce full executable scripts; it only outputs an API name and a few arguments. Together with the cold-start demonstrations, this keeps the "code execution" aspect lightweight and structured, avoiding the difficulty of free-form programming.
> >
> > 4. Tables 1-3 already report the performance of models trained on shuffled patches annotated with sorted IDs, demonstrating the strong effectiveness of AGILE. We additionally trained models directly on datasets where the target is only the sorted ID sequence (without interactive code execution). In this settings, models perform consistently worse than those trained with AGILE's interactive agent loop. This confirms that **the interactive decision-making and perception-driven action sequence**—rather than single-step ID prediction—is essential for transferring perceptual reasoning ability.
> >
> > |                    | MME-RW | RWQA | HRB4K | HRB8K | VStar | MMVP | BLINK | HalBench | MMMU | Avg. |
> > | :----------------: | :----: | :--: | :---: | :---: | :---: | :--: | :---: | :------: | :--: | :--: |
> > |   Qwen2.5-VL-7B    |  44.6  | 68.5 | 68.8  | 65.3  | 76.4  | 74.3 | 56.4  |   50.1   | 54.8 | 62.1 |
> > |       AGILE        |  48.4  | 70.2 | 73.0  | 70.5  | 80.6  | 78.0 | 58.0  |   51.9   | 55.8 | 65.2 |
> > | Without using code |  46.2  | 69.3 | 71.9  | 68.0  | 78.0  | 78.3 | 53.7  |   52.8   | 53.3 | 63.5 |
> >
> > ******************

---

> > > ### Author Response · Authors · 2025-11-19
> > > **Detailed Rebuttal (3/3)**
> > >
> > > > **W4:** Cold-Start only use 1.6K data and RL using 16K data, what if training the model longer during SFT? will the improvements become fewer? Comparisons are needed regarding the scaling of training data.
> > >
> > > We appreciate the reviewer's question regarding the effect of scaling the SFT (cold-start) data. To address this, we expanded the cold-start dataset to **1.5×** and **2×** its original size and conducted additional experiments. The results are as follow:
> > >
> > > |                    | MME-RW | RWQA | HRB4K | HRB8K | VStar | MMVP | BLINK | HalBench | MMMU | Avg. |
> > > | :----------------: | :----: | :--: | :---: | :---: | :---: | :--: | :---: | :------: | :--: | :--: |
> > > |   Qwen2.5-VL-7B    |  44.6  | 68.5 | 68.8  | 65.3  | 76.4  | 74.3 | 56.4  |   50.1   | 54.8 | 62.1 |
> > > | Cold-start (1.6K) |  46.2  | 68.4 | 71.0  | 68.4  | 77.5  | 76.7 | 55.7  |   49.8   | 54.0 | 63.1 |
> > > |        +RL         |  48.4  | 70.2 | 73.0  | 70.5  | 80.6  | 78.0 | 58.0  |   51.9   | 55.8 | 65.2 |
> > > | Cold-start (2.4K) |  46.2  | 68.9 | 71.4  | 68.8  | 77.0  | 76.0 | 57.8  |   50.2   | 54.0 | 63.4 |
> > > |        +RL         |  48.4  | 69.8 | 73.0  | 69.9  | 78.5  | 79.0 | 57.7  |   50.5   | 56.3 | 64.8 |
> > > | Cold-start(3.2K) |  46.5  | 67.8 | 71.4  | 68.8  | 77.0  | 76.7 | 58.1  |   49.2   | 53.3 | 63.2 |
> > > |        +RL         |  47.1  | 70.1 | 73.0  | 70.5  | 79.6  | 78.3 | 57.2  |   50.5   | 55.7 | 64.7 |
> > >
> > > These results show that increasing the SFT data size or training longer during the cold-start stage does not lead to overfitting or any noticeable performance degradation. In practice, the cold-start stage serves a very specific purpose: **teaching the model the tool-use pattern** (i.e., how to invoke the predefined Python APIs). Its goal is not to solve the jigsaw task fully.
> > >
> > > Consequently, the model's performance is largely unaffected by moderate expansions of the SFT set. The primary source of generalization and performance improvement comes from the interactive RL stage, where the model learns to solve jigsaw puzzles through real environment feedback rather than through imitation.
> > >
> > > We have included these additional results and clarification in **Appendix I.5**.
> > >
> > > ******************
> > >
> > > > **Reference**
> > >
> > > [1] DeepEyes: Incentivizing "Thinking with Images" via Reinforcement Learning. Arxiv2505.
> > >
> > > [2] MM-Eureka: Exploring the Frontiers of Multimodal Reasoning with Rule-based Reinforcement Learning. Arxiv2503.

---

> ### Comment · Reviewer_yrdc · 2025-11-25
>
> Looks good, indeed some misunderstanding first time i read the paper, so i will increase my score to 4

---

> ### Author Response · Authors · 2025-11-25
> **Looking for further discussion**
>
> Dear Reviewer yrdc,
>
> Thank you for reviewing our paper and for the valuable feedback you have provided. We greatly appreciate the time and effort you invested in evaluating our work, and your constructive insights have been instrumental in helping us refine the manuscript.
>
> In this rebuttal, we have conducted additional experiments and analyses to thoroughly address the concerns you raised. These include comparisons between AGILE and other reinforcement learning algorithms, analyses of AGILE with and without Python-code action-space design, and extended evaluations of the cold-start phase. We have incorporated all of these updates into the revised version of the paper.
>
> Additionally, we implemented several improvements based on suggestions from the other reviewers, which further enhanced the completeness and solidity of the paper:
>
> 1. Added visual attention analysis comparing AGILE-trained models with baselines. We find that AGILE training leads to more focused and meaningful visual attention on key visual cues. Relative to the initial manuscript, the revised version presents a more comprehensive account of the underlying factors through which AGILE strengthens visual perception and reasoning (Figure 12 and Appendix G).
>
> 2. Added ablation studies on the three reward components (accuracy, format, and step reward). The results show that AGILE is robust to the weighting of accuracy and format terms, while removing the step reward leads to a clear performance drop. This supports our design intuition that the step reward discourages sparse or unfocused reasoning (Appendix I.1).
>
> 3. Extended the training to the more challenging 3 $\times$ 3 setting, indicating that AGILE exhibits favorable scaling properties with respect to both data size and task difficulty, and continues to yield steady improvements as the scale increases (Appendix I.2).
>
> 4. Added ablations on the cold-start phase and full vs. partial action spaces, further illustrating the contribution of each component in AGILE’s design (Appendix I.3).
>
> We are pleased to see that our rebuttal was able to clarify several of the misunderstandings you initially raised. At the same time, we noticed that your current score is 4, which suggests that there may still be some remaining concerns or unresolved questions regarding the paper. If there are any additional issues you would like us to clarify or elaborate on, we would be very grateful if you could let us know. We are more than willing to provide further explanation, supporting evidence, or additional discussion to fully address any doubts.
>
> Your continued engagement would be extremely helpful for us, and we are committed to thoroughly resolving all concerns you might still hold. We remain fully committed to making the work as solid and rigorous as possible and to addressing any remaining concerns you may have.
>
> Sincerely,
> The Authors

---

### Official Review · Reviewer_es5M · 2025-11-01

**Soundness:** 2
**Presentation:** 2
**Contribution:** 2
**Rating:** 2
**Confidence:** 3

**Summary:**

This paper proposes AGILE, an agentic jigsaw interaction learning framework to enhance visual perception and reasoning in VLMs. It formulates jigsaw solving as an interactive environment where the model generates executable Python actions (Swap, Observe, Crop, Zoom), receives fine-grained visual feedback, and iteratively improves through GRPO-based RL. A cold-start stage uses 1.6K expert trajectories (filtered Gemini 2.5 Pro outputs) to teach instruction-following and code generation; RL is then performed on 15.6K programmatically generated 2×2 jigsaw tasks with controllable difficulty and verifiable rewards (accuracy, format, step). Results show large gains on jigsaw puzzles (2×2 Acc: 9.5%→82.8%; Score: 29.4%→89.0%) and nontrivial generalization across 9 vision benchmarks (+3.1% avg), with especially strong improvements on high-resolution understanding. The paper analyzes data scaling, compares jigsaw vs general QA data under a fixed budget, and includes case studies of emergent behaviors.

**Strengths:**

Clear problem framing: shows that current VLMs perform near random on simple jigsaw tasks, motivating a proxy task for fundamental perception and reasoning.

Interactive, verifiable RL: uses executable code and a rule-based environment, enabling precise, fine-grained feedback and scalable supervision without expensive annotation.

Strong empirical gains on target task: very large improvements on 2×2 and meaningful generalization to 3×3 despite training on 2×2.

Cross-task generalization: consistent improvements (+3.1% avg) across diverse downstream benchmarks (HR, real-world, fine-grained, reasoning, hallucination).

Insightful analyses: comparison with General QA data at equal budget; case studies illustrate learned strategies (edge continuity, text alignment, semantic consistency).

**Weaknesses:**

**Limited task significance and generality**

Fixed jigsaw formulation: The current agent–environment interaction is confined to regular grid jigsaws (uniform tiling, no occlusion, no non-rigid deformation, no noise). Such strong scene priors make it hard to cover the diverse layouts and transformations in open-world vision (e.g., free-form crops, affine/perspective distortions, occlusions, and layered structures), limiting the method’s external validity for general visual reasoning.

2×2 performance saturation: Training and reward design are primarily optimized for the 2×2 setting and are near the ceiling (high Acc/Score). There is limited evidence of sustained gains on more complex structures. Overly strong 2×2 metrics may mask real capability gaps on harder tasks.

Weak size generalization: While there is some transfer to 3×3, the accuracy remains low (20.8%). Moreover, larger grids (3×3, 4×4) are not directly trained in the RL stage. The absence of cross-size curricula or memory mechanisms leads to insufficient generalization from small to larger scales.

Fixed, non-general action space: The toolset of Swap/Observe/Crop/Zoom is highly tailored to jigsaws and does not seamlessly transfer to general multimodal interactions (e.g., instance-level operations, relation editing, cross-image search). There is a lack of validation and ablation on extensible action semantics (parameterized operations, composable tools).


**Uncertain sources of gains on general tasks**

Improvements may stem more from distillation than from the jigsaw task: The cold-start phase relies on Gemini 2.5 Pro–generated and filtered expert trajectories, which may impart strong priors to the base model in instruction-following, code formatting, and visual alignment. Current controls are insufficient to disentangle the “expert distillation effect” (SFT-driven general gains) from the independent contribution of “jigsaw agentic interaction + RL.” Suggested additions:
RL-only (without expert trajectories) or weak-expert/random-teacher controls;
Controls with distillation from general multimodal tasks at equal scale;
Ablations removing operations tightly tied to expert demonstrations (e.g., Crop/Zoom) to see whether downstream gains persist.

Coupling risk in rewards/training signals: Format rewards and step penalties may encourage “normalized outputs and low-step strategies” rather than genuine perception–reasoning improvements. It is necessary to verify whether the gains reflect core abilities rather than policy biases through analyses of failure types (code errors vs. cognitive errors) and cross-task robustness under occlusion, distortion, and noise.

**Questions:**

See weakness.

---

> ### Author Response · Authors · 2025-11-19
> **Detailed Rebuttal (1/6)**
>
> We sincerely thank you for your constructive feedback and valuable suggestions. Below we address the raised concerns point-by-point and provide additional analyses and clarifications.
>
> > **W1-1:** Fixed jigsaw formulation: The current agent-environment interaction is confined to regular grid jigsaws (uniform tiling, no occlusion, no non-rigid deformation, no noise). Such strong scene priors make it hard to cover the diverse layouts and transformations in open-world vision (e.g., free-form crops, affine/perspective distortions, occlusions, and layered structures), limiting the method's external validity for general visual reasoning.
>
> Thank you for raising this concern. We would like to emphasize that, despite its seemingly simple structure, the proposed jigsaw environment is far from trivial from a VLM's perceptual-reasoning perspective.
>
> 1. Even in the "regular grid" setting, current frontier VLMs fail dramatically, showing that this task is fundamentally hard, not simplistic. As shown in **Table 1**, even flagship models such as GPT-4o, Gemini-2.5-Pro, and Qwen2.5-VL-72B struggle on the 2 $\times$ 2 jigsaw benchmark. This reveals that basic tile-level perceptual consistency, fine-grained spatial alignment, and local-to-global reasoning **remain unsolved capabilities for today's VLMs**, even without non-rigid deformation, occlusion, or perspective distortion. In other words, VLMs already fail before any open-world complexity is introduced. This makes the task a meaningful diagnostic and training signal, rather than an oversimplified setting.
>
> 2. The goal of this work is not to simulate the diverse layouts and transformations of the open world, but rather to construct a **scalable, verifiable, low-cost proxy task** that enhances the foundational **perception and reasoning** of VLMs by providing strong visual self-supervision signals. This is a critical aspect that current multimodal reinforcement learning pipelines, particularly on the data side. This design follows the same logic as recent work in agentic reinforcement learning tasks (e.g., Logic-RL's Knights & Knaves puzzle [1], Enigmata [2], Game-RL [3], etc.), where the **structure of the proxy task** and **the behavioral patterns it elicits in models**, rather than its surface similarity to open-world environments, is key to providing transferable reasoning improvements.
>
> 3. Methodologically, our AGILE framework is decoupled from the image content itself. Affine distortions, occlusions, noise, and free-form crops can all be incorporated during the data generation phase without modifying our reinforcement learning framework or agent design. In fact, our RL dataset already includes **high-resolution, noisy, cluttered real-world scenes** (e.g., **RealWorldQA**, **HRBench**, **BLINK**), and the model has consistently demonstrated performance improvements on these benchmarks. Therefore, the **jigsaw proxy task** is a means to cultivate perceptual reasoning abilities, rather than simulate all real-world transformations.
>
> ***************
>
> > **W1-2:** 2 $\times$ 2 performance saturation: Training and reward design are primarily optimized for the 2 $\times$ 2 setting and are near the ceiling (high Acc/Score). There is limited evidence of sustained gains on more complex structures. Overly strong 2 $\times$ 2 metrics may mask real capability gaps on harder tasks.
>
> We respectfully note that this comment may be based on a misunderstanding of the 2 $\times$ 2 jigsaw task, and we would like to clarify as follows.
>
> 1. **2 $\times$ 2 has exposed significant capability deficiencies in existing models.** Baseline VLMs already perform poorly on 2 $\times$ 2 tasks, e.g., **Qwen2.5-VL-7B**: accuracy **9.5%**, **Qwen2.5-VL-72B**: accuracy **27.4%**. This indicates that the **2 $\times$ 2 jigsaw exposes a fundamental deficiency** in basic tile-level consistency understanding, rather than being a trivial or saturated task.
>
> 2. **The scaling law curve (Fig. 3 Impact of Training Data Scale) shows that performance has not saturated.** As reinforcement learning data increases, both **jigsaw accuracy** and **general task performance** (e.g., RWQA, HRBench) show clear upward trends. There is no sign of overfitting, and performance improvements do not stagnate prematurely. We continuously monitor the **validation curve** during training to determine the appropriate amount of RL data, preventing excessive reinforcement. Therefore, "saturation" is not a true phenomenon.
>
> 3. **2 $\times$ 2 is a capability-matched starting point.** Larger grids would significantly increase both the context length and combinatorial complexity of the jigsaw. Starting with 2 $\times$ 2 avoids early-stage learning difficulties in reinforcement learning, aligning with **curriculum learning** principles.
>
> ***************

---

> > ### Author Response · Authors · 2025-11-19
> > **Detailed Rebuttal (2/6)**
> >
> > > **W1-3:** Weak size generalization: While there is some transfer to 3 $\times$ 3, the accuracy remains low (20.8%). Moreover, larger grids (3 $\times$ 3, 4 $\times$ 4) are not directly trained in the RL stage. The absence of cross-size curricula or memory mechanisms leads to insufficient generalization from small to larger scales.
> >
> > We fully agree that larger grids are an important direction for future work, and would like to further clarify the following points:
> >
> > **The difficulty of 3 $\times$ 3 puzzles is much higher than that of 2 $\times$ 2**. Although a 3 $\times$ 3 jigsaw puzzle appears simple to humans, it constitutes a high-complexity combinatorial search problem for VLM-based agents, far exceeding the effective reasoning horizon of even state-of-the-art models. A 3 $\times$ 3 puzzle contains 9 tiles, resulting in a configuration space of 9! = 362,880. This is over **15,000$\times$ larger** than the 2 $\times$ 2 case (4! = 24). Thus, the agent must identify a single correct arrangement from hundreds of thousands of possibilities. This transforms the task into a large discrete reasoning problem rather than a conventional perception or VQA task. Each swap introduces substantial branching in the search space, making single-step visual cues insufficient. As shown in our experiments (Table 1: Jigsaw Accuracy), all existing VLMs exhibit extremely low performance on 3 $\times$ 3 puzzles, highlighting the intrinsic difficulty of this setting.
> >
> > Thus, for Qwen2.5-VL-7B, whose initial accuracy is merely **0.4%**, achieving **20.8%** on 3 $\times$ 3 puzzles after training only on 2 $\times$ 2 tasks represents a substantial improvement, far surpassing the performance of closed-source and LLM-scale VLMs like **GPT-4o**, **Gemini-2.5-Pro**, and **Qwen2.5-VL-72B**.
> >
> > To further address your concern, we add an additional 8K RL training stage on 3 $\times$ 3 puzzles after the 2 $\times$ 2 training. The results show that the model trained on the 3 $\times$ 3 setting achieves additional performance gains on downstream tasks.
> >
> > |                    | MME-RW | RWQA | HRB4K | HRB8K | VStar | MMVP | BLINK | HalBench | MMMU | Avg. |
> > | :----------------: | :----: | :--: | :---: | :---: | :---: | :--: | :---: | :------: | :--: | :--: |
> > |   Qwen2.5-VL-7B    |  44.6  | 68.5 | 68.8  | 65.3  | 76.4  | 74.3 | 56.4  |   50.1   | 54.8 | 62.1 |
> > | 15.6K 2 $\times$ 2 |  48.4  | 70.2 | 73.0  | 70.5  | 80.6  | 78.0 | 58.0  |   51.9   | 55.8 | 65.2 |
> > | 15.6K 2 $\times$ 2 + 8K 3 $\times$ 3 |  50.4  | 70.6 | 73.4  | 70.5  | 80.1  | 79.0 | 58.1  |   52.1   | 55.8 | 65.6 |
> >
> > It is important to note that **AGILE's methodology is extendable to larger grids**. However, given the limited initial capabilities of current models, the context length and number of exploratory steps needed to solve larger grids grow non-linearly as the number of tiles increases. Therefore, we chose to begin with **a difficulty level that matches the model's capabilities** (a core principle of curriculum learning in RL). Crucially, this also highlights the **sustainability** of our method. As base VLMs continue to advance in perceptual and reasoning capacity, AGILE's controllable-difficulty framework can seamlessly scale to larger grid sizes, ensuring that the framework remains effective over time.
> >
> > The above results and analyses have been incorporated into **Appendix I.2**, providing additional evidence for the effectiveness of AGILE.
> >
> >
> > ***************

---

> > > ### Author Response · Authors · 2025-11-19
> > > **Detailed Rebuttal (3/6)**
> > >
> > > > **W1-4:** Fixed, non-general action space: The toolset of Swap/Observe/Crop/Zoom is highly tailored to jigsaws and does not seamlessly transfer to general multimodal interactions (e.g., instance-level operations, relation editing, cross-image search). There is a lack of validation and ablation on extensible action semantics (parameterized operations, composable tools).
> > >
> > > While we appreciate the valuable comment regarding the generalization of the action space, we would like to clarify that the **basic perceptual actions** used in AGILE (**cropping/zooming/observing**) are not specific to the jigsaw setting. Instead, they have been widely applied as **core visual exploration primitives** in other multimodal agent systems (e.g., OpenAI's Think-with-Image (o3) [4]; visual document agents such as VRAG-RL [5]; and multi-step visual tool-use models such as DeepEyes [6]). These studies consistently show that the above **perceptual actions** can enhance **general visual perception and reasoning ability**, rather than being limited to any specific downstream task. Building on this insight, AGILE establishes a concise perceptual action space that effectively **unlocks the model's potential for fine-grained perception and reasoning** through the jigsaw proxy task.
> > >
> > > Following your suggestion, we also validate that our action space is **not limited** to the jigsaw task by evaluating the AGILE-trained model on general multimodal benchmarks using an interaction paradigm similar to **o3 (Think-with-Image)**, which involves task patterns entirely unseen during training (e.g., cropping/zooming/**rotating images, drawing auxiliary lines, creating Set-of-Marks (SOM), and performing code-based calculations**). We compare Qwen2.5-VL-7B with its counterpart trained on 15.6K interactive jigsaw examples, where both models are instructed to **use the above tools and actions during downstream inference**.
> > >
> > > |               | MME-RW | HRB4K | HRB8K | MMStar | MathVista | WeMath | MMMU | Avg. |
> > > | :-----------: | :----: | :---: | :---: | :----: | :-------: | :----: | :--: | :--: |
> > > | Qwen2.5-VL-7B |  44.8  | 63.0  | 53.4  |  59.1  |   62.0    |  31.0  | 47.1 | 51.5 |
> > > | Cold-Start+RL |  49.4  | 72.3  | 67.3  |  60.5  |   63.2    |  34.3  | 51.3 | 56.9 |
> > >
> > > The experimental results confirm that the action space acquired through AGILE's agentic training on the jigsaw task **transfers seamlessly to broader multimodal tasks** (for example, using auxiliary lines to enhance reasoning in mathematical scenarios or employing code for computational assistance) .
> > >
> > > The examples raised by the reviewer **(instance-level operations, relation editing, cross-image search, and fully composable toolchains)** indeed represent broader categories of multimodal agent workflows, and we agree that exploring such high-level, task-specific tools is an important direction for future work. At the same time, we believe the actions mentioned by the reviewer are not in conflict with AGILE. Benefiting from the **modularity** of our agent-environment loop and learning algorithm, our framework readily supports the integration of richer action semantics. However, AGILE's objective is **not to exhaust all possible interaction paradigms**, but rather to establish a **scalable, well-controlled proxy task** that can reliably strengthen **end-to-end perceptual reasoning** in VLMs under a practical RL budget.
> > >
> > > ***************

---

> > > > ### Author Response · Authors · 2025-11-19
> > > > **Detailed Rebuttal (4/6)**
> > > >
> > > > > **W2-1:** Improvements may stem more from distillation than from the jigsaw task: The cold-start phase relies on Gemini 2.5 Pro-generated and filtered expert trajectories, which may impart strong priors to the base model in instruction-following, code formatting, and visual alignment. Current controls are insufficient to disentangle the "expert distillation effect" (SFT-driven general gains) from the independent contribution of "jigsaw agentic interaction + RL." Suggested additions: RL-only (without expert trajectories) or weak-expert/random-teacher controls; Controls with distillation from general multimodal tasks at equal scale; Ablations removing operations tightly tied to expert demonstrations (e.g., Crop/Zoom) to see whether downstream gains persist.
> > > >
> > > > To further disentangle the effect of Gemini 2.5 Pro trajectories and the contribution of crop/zoom operations, we conducted an additional experiment on Qwen2.5-VL-7B, where we performed 15.6K jigsaw RL training **without using any expert trajectories**. We included two conditions: (1) RL with the full action space, and (2) RL with **crop/zoom removed**. The results are as follows:
> > > >
> > > > |                             | MME-RW | RWQA | HRB4K | HRB8K | VStar | MMVP | BLINK | HalBench | MMMU | Avg. |
> > > > | :-------------------------: | :----: | :--: | :---: | :---: | :---: | :--: | :---: | :------: | :--: | :--: |
> > > > |        Qwen2.5-VL-7B        |  44.6  | 68.5 | 68.8  | 65.3  | 76.4  | 74.3 | 56.4  |   50.1   | 54.8 | 62.1 |
> > > > |         Cold-Start          |  46.2  | 68.4 | 71.0  | 68.4  | 77.5  | 76.7 | 55.7  |   49.8   | 54.0 | 63.1 |
> > > > |        Cold-Start+RL        |  48.4  | 70.2 | 73.0  | 70.5  | 80.6  | 78.0 | 58.0  |   51.9   | 55.8 | 65.2 |
> > > > | Only RL (full action space) |  46.3  | 69.5 | 71.5  | 68.1  | 78.5  | 79.0 | 56.3  |   53.2   | 53.1 | 63.9 |
> > > > | Only RL (remove crop/zoom)  |  46.2  | 69.3 | 71.9  | 68.0  | 78.0  | 78.3 | 53.7  |   52.8   | 53.3 | 63.5 |
> > > >
> > > > Even **without any Gemini 2.5 Pro trajectory supervision**, direct RL training still yields a **+1.8% improvement** over the original Qwen2.5-VL-7B baseline (**63.9% vs. 62.1%**). This demonstrates that the performance gains indeed come from the **jigsaw agentic interaction + RL**, rather than from knowledge distillation in the cold-start phase. Moreover, removing the **crop/zoom** operations leads to a noticeable drop in performance (particularly on detail-sensitive benchmarks such as BLINK, where performance decreases from **56.3% to 53.7%**), further confirming the **necessity of the AGILE interaction design** for enabling effective perceptual reasoning improvements.
> > > >
> > > > We have incorporated these additional experiments and analyses into **Appendix I.3**, so that readers can more clearly understand the contributions of different training stages and action-space designs to the model's capability gains.
> > > > ***************

---

> > > > > ### Author Response · Authors · 2025-11-19
> > > > > **Detailed Rebuttal (5/6)**
> > > > >
> > > > > > **W2-2:** Coupling risk in rewards/training signals: Format rewards and step penalties may encourage "normalized outputs and low-step strategies" rather than genuine perception-reasoning improvements. It is necessary to verify whether the gains reflect core abilities rather than policy biases through analyses of failure types (code errors vs. cognitive errors) and cross-task robustness under occlusion, distortion, and noise.
> > > > >
> > > > > Thank you for raising this important issue.
> > > > > In AGILE's reward design, the format reward and step penalty serve as stabilizing mechanisms. The format reward simply ensures that the code is parseable, without providing any visual shortcuts. The step penalty is adaptively adjusted according to the model's capability stage. In the early stages of training, the model's perceptual ability is weak, and it naturally requires more interaction steps to inspect tiles, zoom in on them, and gradually learn the basic concept of tile-level consistency. At this stage, to avoid incurring the maximum step penalty for incorrect reasoning, the model voluntarily performs more exploratory actions, which improves its task accuracy. As training progresses, the model develops denser and more reliable perceptual reasoning capabilities, and it naturally converges to shorter trajectories. This shift occurs **not due to a preference for fewer steps**, but because fewer exploratory actions are needed to achieve the correct solution. Thus, the same reward mechanism effectively supports early-stage exploration and later-stage efficient reasoning, rather than enforcing a fixed behavioral preference, as we discuss in the **wandb optimization curves** analysis in **Appendix E**.
> > > > >
> > > > > To further validate the effectiveness of the step penalty, we conducted an ablation study. As shown below, removing the penalty results in a consistent drop in general-purpose downstream benchmarks (-0.8% on average), further demonstrating that the step penalty contributes to strengthening the model's perceptual and reasoning abilities.
> > > > > |                    | MME-RW | RWQA | HRB4K | HRB8K | VStar | MMVP | BLINK | HalBench | MMMU | Avg. |
> > > > > | :----------------: | :----: | :--: | :---: | :---: | :---: | :--: | :---: | :------: | :--: | :--: |
> > > > > |   Qwen2.5-VL-7B    |  44.6  | 68.5 | 68.8  | 65.3  | 76.4  | 74.3 | 56.4  |   50.1   | 54.8 | 62.1 |
> > > > > | **w** step reward  |  48.4  | 70.2 | 73.0  | 70.5  | 80.6  | 78.0 | 58.0  |   51.9   | 55.8 | 65.2 |
> > > > > | **wo** step reward |  47.4  | 71.1 | 72.4  | 69.8  | 78.5  | 75.3 | 56.7  |   52.8   | 55.6 | 64.4 |
> > > > >
> > > > >
> > > > > Beyond quantitative ablations, we also conducted a failure-type analysis to verify that the performance improvements are not the result of policy bias. In the early stages of RL, **code-format errors** and **cognitive errors** accounted for **10.6%** and **73.0%**, respectively. After optimizing for 200 steps, **code-format errors** and **cognitive errors** decreased to approximately **0%** and **17.2%**, respectively. The reduction in **cognitive errors** (misalignment, visual misunderstanding) is much greater than the reduction in **code-format or execution errors**, indicating that the model's performance improvements stem from stronger **perceptual reasoning capabilities**, not from policy biases.
> > > > >
> > > > > Moreover, AGILE has demonstrated **consistent improvements** across task benchmarks involving **occlusion, distortion, clutter, and noise** (e.g., **RealWorldQA**, **HRBench**, **BLINK**) as shown in **Table 3: Main Results** of the paper. Since we employ the **exact same evaluation methods** for general downstream benchmarks as those used for the baseline (the same prompts, and following the extraction and matching principles of **VLMEvalKit**), there are no concerns regarding step counts or output formats.
> > > > >
> > > > > Finally, we performed extra **qualitative analysis** on representative cases in the downstream benchmark tasks, with a particular focus on **attention map visualizations**, as attention patterns provide insight into the underlying behavior of the model. We compared the **visual attention distributions** when responding to general downstream tasks between the original **Qwen2.5-VL-7B** model and the **AGILE-trained model**. We observed that **Qwen2.5-VL-7B** exhibited dispersed and inconsistent attention, especially in high-resolution scenes or tasks requiring fine-grained visual cues. This insufficient attention to critical visual evidence limited its perceptual accuracy and reasoning efficiency. In contrast, the **AGILE-trained model** consistently allocated highly concentrated attention to key visual tokens, enabling it to extract details more accurately and perform more reliable visual reasoning in multimodal tasks.

---

> > > > > > ### Author Response · Authors · 2025-11-19
> > > > > > **Detailed Rebuttal (6/6)**
> > > > > >
> > > > > > These results indicate that the performance improvements reflect a **genuine enhancement of visual reasoning abilities**, rather than policy biases caused by the reward mechanism. We have incorporated **the ablations of step penalty** (**Table 6**, **Appendix I.1**) and these **qualitative case studies and visualizations** (**Figure 12**, **Appendix G**) in the revised manuscript to more clearly and substantively demonstrate that the AGILE methodology can indeed improve **perceptual and reasoning capabilities**.
> > > > > >
> > > > > > > **Reference**
> > > > > >
> > > > > > [1] Logic-RL: Unleashing LLM Reasoning with Rule-Based Reinforcement Learning. Arxiv2502.
> > > > > >
> > > > > > [2] Enigmata: Scaling Logical Reasoning in Large Language Models with Synthetic Verifiable Puzzles. Arxiv2505.
> > > > > >
> > > > > > [3] Game-RL: Synthesizing Multimodal Verifiable Game Data to Boost VLMs' General Reasoning. Arxiv2505.
> > > > > >
> > > > > > [4] OpenAI. Thinking with images.
> > > > > >
> > > > > > [5] VRAG-RL: Empower Vision-Perception-Based RAG for Visually Rich Information Understanding via Iterative Reasoning with Reinforcement Learning. NeurIPS2025.
> > > > > >
> > > > > > [6] DeepEyes: Incentivizing "Thinking with Images" via Reinforcement Learning. Arxiv2505.

---

> > > > > > > ### Author Response · Authors · 2025-11-27
> > > > > > > **Welcome for Further Feedback**
> > > > > > >
> > > > > > > Dear Reviewer es5M,
> > > > > > >
> > > > > > > We sincerely thank you for the constructive feedback provided during the reviewing phase. Following your comments, we have conducted substantial additional analyses, new experiments, and extended clarifications throughout our detailed rebuttal (Sections 1–6). The improvements and newly added results directly address all raised concerns:
> > > > > > >
> > > > > > > - **Jigsaw formulation & generality (W1-1):**
> > > > > > >   We expanded the discussion to clarify why even regular-grid jigsaws remain fundamentally challenging for current frontier VLMs, and why the proxy task meaningfully contributes to transferable perceptual reasoning. We also emphasized that AGILE is decoupled from image content and can naturally incorporate distortions, occlusion, or free-form crops.
> > > > > > > - **2 $\times$ 2 saturation & scaling behavior (W1-2):**
> > > > > > >   We provided scaling curve evidence showing non-saturation, clarified the role of curriculum-aligned difficulty, and demonstrated that 2 $\times$ 2 remains far from trivial for existing VLMs.
> > > > > > > - **On size generalization (W1-3):**
> > > > > > >   We added a new RL stage on 3 $\times$ 3 puzzles (+8K examples) and reported improved downstream performance (Appendix. I.2), verifying AGILE's scalability.
> > > > > > > - **Action-space generality (W1-4):**
> > > > > > >   We added new evaluations under Think-with-Image–style interactive settings, showing that the learned perceptual actions transfer to unseen multimodal reasoning tasks.
> > > > > > > - **On disentangling distillation vs. RL (W2-1):**
> > > > > > >   We conducted **new ablations**: RL-only (without any expert trajectories) and RL-only with/without crop/zoom. Results show clear improvements even without cold-start distillation, confirming the independent contribution of AGILE's agentic interaction.
> > > > > > > - **Reward-coupling risks & policy bias (W2-2):**
> > > > > > >   We added step-penalty ablations (Table 6, Appendix. I.1) and failure-type analyses, showing cognitive-error reduction rather than format-based behavioral bias. We also included qualitative attention-visualization studies (Appendix. G).
> > > > > > >
> > > > > > > Across all concerns, we have incorporated the expanded analyses, new experiments, and clarifications into the revised manuscript and appendix for transparency and completeness.
> > > > > > >
> > > > > > > We would like to kindly ask whether our responses and additional results have sufficiently addressed your concerns, and whether you might consider updating your evaluation or final rating accordingly. We truly appreciate the time and expertise you have devoted to reviewing our work and would be grateful for any further feedback you may have.
> > > > > > >
> > > > > > > Best regards,
> > > > > > > The Authors

---

### Official Review · Reviewer_ciSx · 2025-11-01

**Soundness:** 3
**Presentation:** 3
**Contribution:** 3
**Rating:** 6
**Confidence:** 4

**Summary:**

This paper presents AGILE, a novel framework that enhances the visual perception and reasoning capabilities of Vision-Language Models through interactive jigsaw puzzle solving. AGILE formulates jigsaw solving as an iterative interaction between the model and its environment, allowing the model to progressively refine its perceptual and reasoning skills through exploration and feedback. The framework employs a scalable data generation pipeline to construct high-quality multimodal reinforcement learning datasets with controllable difficulty, effectively addressing the scarcity of such data. Experimental results show that AGILE significantly improves jigsaw-solving performance and generalizes well across diverse vision tasks, demonstrating its strong potential for advancing the development of VLMs.

**Strengths:**

1. Novel Framework: The proposed AGILE framework creatively formulates jigsaw solving as an interactive process to enhance VLMs’ perceptual and reasoning abilities. This integration of interactive learning with jigsaw-based perception is highly novel and opens up a promising research direction for improving the core competencies of VLMs, particularly within self-supervised learning paradigms.
2. Significant Performance Gains: Experimental results show that models trained with AGILE achieve substantial improvements on jigsaw tasks and exhibit strong generalization across various vision benchmarks, clearly demonstrating the framework’s effectiveness in boosting perception and reasoning.
3. Data Generation Advantage: The proposed scalable data generation method is of notable value, as it effectively addresses the data scarcity challenge in multimodal reinforcement learning by producing high-quality, controllable training datasets.
4. The paper is clearly written, logically organized, and easy to follow.

**Weaknesses:**

1. It remains unclear whether AGILE adopts a multi-turn or single-turn evaluation setup for general downstream tasks. The paper only mentions that both AGILE and the baselines use VLMEvalKit as the evaluation framework, but provides no further details about the evaluation protocol. If AGILE employs multi-turn interactions while the baselines use single-turn ones, this may raise concerns about the fairness of the comparison.
2. The motivation behind the Step Reward design is not well explained. It would be helpful to understand how removing this component affects training dynamics, especially its impact on general downstream vision tasks. Including ablation experiments on this aspect would strengthen the paper’s methodological soundness and justification.
3. Since AGILE is based on multi-turn interactive reinforcement learning, it likely introduces additional training overhead compared with single-turn RL. The authors are encouraged to discuss the computational cost and efficiency trade-offs. A comparison between single-turn and multi-turn settings in terms of both performance and training cost would make the paper more comprehensive.

**Questions:**

1. Could the authors clarify whether AGILE uses a multi-turn or single-turn setting during evaluation on general downstream tasks? If multi-turn interactions are involved, how is fairness ensured when comparing with single-turn baselines?
2. What is the underlying motivation for introducing the Step Reward design? Could the authors elaborate on how this component influences the training dynamics and model performance? It would be valuable to include an ablation study to show the effect of removing this reward, particularly on general vision tasks.
3. Since AGILE involves multi-turn reinforcement learning, could the authors provide a quantitative or qualitative analysis of the additional computational cost introduced by this setup? A comparison between single-turn and multi-turn RL in terms of performance–cost trade-offs would help clarify the practical implications of the proposed method.

---

> ### Author Response · Authors · 2025-11-19
> **Detailed Rebuttal (1/2)**
>
> We appreciate your recognition of our work as well as your insightful comments and thoughtful suggestions. Below, we address each concern in detail and provide further analyses and clarifications.
>
> > **W1&Q1:** Could the authors clarify whether AGILE uses a multi-turn or single-turn setting during evaluation on general downstream tasks? If multi-turn interactions are involved, how is fairness ensured when comparing with single-turn baselines?
>
> We clarify that all general downstream evaluations are **strictly conducted in single-turn mode, fully aligning with the standard VLMEvalKit framework**. AGILE does **not** employ multi-turn interactions during inference on general benchmarks. Multi-turn reasoning is used only during RL training, where the model interacts with the jigsaw environment. For all downstream tasks, the AGILE-trained model is evaluated **with the same single-turn prompts as all baselines** (Qwen2.5-VL-7B, InternVL, LLaVA-OV, etc.).
>
> We explicitly state this in **Sec.4.1** to avoid confusion and reinforce that the comparison is entirely fair and standardized.
>
> > **W2&Q2:** What is the underlying motivation for introducing the Step Reward design? Could the authors elaborate on how this component influences the training dynamics and model performance? It would be valuable to include an ablation study to show the effect of removing this reward, particularly on general vision tasks.
>
> We thank the reviewer for raising this point.
>
> The step reward is designed to encourage the model to reason concisely and efficiently while correctly completing the task, thereby avoiding unnecessary or redundant interactions. Our observations show that excessive steps typically correspond to **sparse or unfocused visual reasoning**, whereas completing the task in fewer steps leads to **more sufficient and well-structured single-step reasoning**. Guided by the step reward, the model initially relies on multiple interactions to gather sufficient information and avoid mistakes that would trigger the maximum-step penalty. As its capability improves, the model naturally reduces redundant interactions, eventually reaching a point where each single-step reasoning contains all the essential information required for task completion.
>
> To substantiate this, we conducted additional ablation experiments. Removing the step reward results in a clear performance drop on general downstream tasks (approximately -0.8%). These findings confirm that the step-penalty design is both beneficial and essential for guiding effective learning.
>
> |                    | MME-RW | RWQA | HRB4K | HRB8K | VStar | MMVP | BLINK | HalBench | MMMU | Avg. |
> | :----------------: | :----: | :--: | :---: | :---: | :---: | :--: | :---: | :------: | :--: | :--: |
> |   Qwen2.5-VL-7B    |  44.6  | 68.5 | 68.8  | 65.3  | 76.4  | 74.3 | 56.4  |   50.1   | 54.8 | 62.1 |
> | **w** step reward  |  48.4  | 70.2 | 73.0  | 70.5  | 80.6  | 78.0 | 58.0  |   51.9   | 55.8 | 65.2 |
> | **wo** step reward |  47.4  | 71.1 | 72.4  | 69.8  | 78.5  | 75.3 | 56.7  |   52.8   | 55.6 | 64.4 |
>
> We have added a more detailed analysis and discussion of the step-reward ablation in **Appendix I.1** of the revised manuscript.

---

> > ### Author Response · Authors · 2025-11-19
> > **Detailed Rebuttal (2/2)**
> >
> > > **W3&Q3:** Since AGILE involves multi-turn reinforcement learning, could the authors provide a quantitative or qualitative analysis of the additional computational cost introduced by this setup? A comparison between single-turn and multi-turn RL in terms of performance-cost trade-offs would help clarify the practical implications of the proposed method.
> >
> > We appreciate the reviewer's suggestion to discuss the efficiency trade-offs of AGILE, as computational cost is an important consideration for both the research community and the reproducibility of our training frameworks.
> >
> > While AGILE's multi-turn reinforcement learning inevitably incurs additional computational overhead from environment interactions, the overall training cost remains well within a manageable range. The average rollout length over the entire training process is approximately 1.58 steps, indicating that multi-turn RL introduces only modest additional cost compared to single-turn RL. Importantly, a core motivation behind AGILE is to **formulate the jigsaw task as an agent-environment interaction problem**, thereby simplifying the learning difficulty in accordance with curriculum learning principles. We observe that the base model (Qwen2.5-VL-7B) exhibits **very limited initial capability**, making it extremely challenging to complete a jigsaw in a single shot (as shown in Table 1: Jigsaw Accuracy). Training the model to perform the entire task at once in the early stage provides **little meaningful learning signal**. Therefore, decomposing a complex task into a sequence of sub-actions (e.g., *swap*, *observe*, etc.) is necessary. In the early phase, models with limited capabilities rely on a greater number of swaps and observations, while the required number of tool calls decreases as their abilities improve. This design provides a **progressive learning pathway** for solving jigsaw tasks and makes the RL optimization process substantially easier.
> >
> > We also compare AGILE with a concurrent work [1]. As shown in the table, AGILE achieves **comparable improvements** while using **significantly fewer training steps, smaller batch sizes, and fewer rollouts**, highlighting the efficiency and effectiveness of our approach.
> >
> > |                   | Batch Size | Train Step | Rollout N | Rollout Turn |
> > | ----------------- | ---------- | ---------- | --------- | ------------ |
> > | Visual jigsaw [1] | 256        | 1000       | 16        | 1            |
> > | Ours              | 64         | 244        | 8         | 1.58 (average)|
> >
> > |                   | MMVP     | HRB8K | Vstar    | MME-RW   | Avg.     |
> > | ----------------- | -------- | ----- | -------- | -------- | -------- |
> > | Visual jigsaw [1] | **+6.0** | +3.8  | +3.7     | +2.6     | +4.0     |
> > | Ours              | +3.7     | **+5.2**  | **+4.2** | **+3.8** | **+4.2** |
> >
> >
> >
> > > **Reference**
> >
> > [1] Visual Jigsaw Post-Training Improves MLLMs. Arxiv2509.

---

> ### Comment · Reviewer_ciSx · 2025-11-26
>
> Thanks for your detailed reply. All of my concerns have been well addressed.
>
> The paper provides a scalable approach for VLM training and further validates its practicality on extensive downstream tasks. Given the method's universality and general applicability, I raise my score to 8 and advocate the acceptance for this paper.

---

> > ### Author Response · Authors · 2025-11-26
> >
> > Dear Reviewer ciSx,
> >
> > Thank you very much for your recognition of our responses. We are glad to hear your positive feedback, and we sincerely appreciate it. Following your constructive suggestions, we will continue to refine and improve the paper.
> >
> > Many thanks again for your thoughtful comments, time, and patience.
> >
> > Sincerely,
> > The Authors

---

### Official Review · Reviewer_Jwkt · 2025-11-03

**Soundness:** 3
**Presentation:** 3
**Contribution:** 3
**Rating:** 6
**Confidence:** 3

**Summary:**

This paper proposes AGILE (Agentic Jigsaw Interaction Learning), a framework to improve visual perception and reasoning in large Vision-Language Models (VLMs). The authors propose a RL based paradigm where the model progressively solves visual jigsaw puzzles. AGILE formulates the puzzle as interaction between the model and an external environment, with the model generating executable Python code to perform actions such as swapping, cropping, or zooming image tiles. Through iterative observation and feedback, the model develops stronger reasoning and perception capabilities. The framework includes a cold-start phase, leveraging expert trajectories collected via Gemini 2.5 Pro, followed by a reinforcement phase using Group Relative Policy Optimization (GRPO). Extensive experiments demonstrate accuracy improvement on the trained task from 9.5% to 82.8% on 2×2 puzzles. The trained model also demonstrates generalization gains (average +3.1%) across nine diverse vision benchmarks. By constructing scalable, controllable jigsaw datasets without reliance on costly human annotation, AGILE provides a scalable solution to the scarcity of high-quality multimodal RL data.

**Strengths:**

The paper’s key strength lies in using jigsaw solving as an interactive proxy task for improving visual reasoning. Because of the rule-based data generation, this method prevents the need of annotations in a scalable manner. This reduces dataset scarcity in multimodal reinforcement learning. This design also enables control over task difficulty.

By integrating agentic behavior, code generation, and environmental feedback, the authors create a setting that closely simulates exploratory problem-solving. This becomes an essential step toward autonomous multimodal reasoning.

Although some gains on the trained jigsaw task are expected over the base model, using AGILE achieves a jump from 9.5% to 82.8%. The model trained using AGILE also achieves gains on the general vision tasks.

Experiments on the RL training data scale also indicate the importance of the RL step in improving model performance. Further experiments on using a mix of General QA and Jigsaw data over only General QA data to improve the performance indicating the importance of the jigsaw data.

**Weaknesses:**

- While the authors report an average 3.1% generalization improvements, it is not clear whether these improvements are mere metric increases, or because of an improvement in visual perception and reasoning. While the authors provide a small case study on the jigsaw task, qualitative analysis on the general benchmarks would be useful to quantify the gains.
- The reliance on python code generation for performing actions can limit the model capabilities for further generalization to unstructured tasks. The authors do not mention/explore the use of more traditional tool calling methods to directly call environment APIs instead of generating python code to execute the APIs.

**Questions:**

- Since the model generates Python code to perform jigsaw actions, how are code generation errors handled during RL training? Are invalid executions penalized, or filtered out before reward calculation?
- The paper defines three components (accuracy, format, and step rewards). Could the authors clarify how sensitive overall performance is to the weighting coefficients (α, β, γ)? Have ablation tests been performed to determine the optimal balance among these terms?
- The model shows improvement across nine benchmarks. Could the authors elaborate on which specific reasoning categories (e.g., fine-grained recognition vs. scene reasoning) benefit most from AGILE’s training?

---

> ### Author Response · Authors · 2025-11-19
> **Detailed Rebuttal (1/3)**
>
> We sincerely thank you for your constructive feedback and valuable suggestions. Below we address the raised concerns point-by-point and provide additional analyses and clarifications.
>
> > **W1:** While the authors report an average 3.1% generalization improvements, it is not clear whether these improvements are mere metric increases, or because of an improvement in visual perception and reasoning. While the authors provide a small case study on the jigsaw task, qualitative analysis on the general benchmarks would be useful to quantify the gains.
>
> Thank you for this insightful comment, which motivates us to further investigate the substantive mechanisms through which AGILE enhances the model's perceptual and reasoning abilities, rather than focusing on its improvements in evaluation scores.
>
> To address this point, we conduct additional qualitative analyses on representative cases from downstream benchmarks, focusing in particular on **attention-map visualizations**, as attention patterns offer an intrinsic view into a model's underlying behavior. Specifically, we compare the visual-token attention distributions of the original Qwen2.5-VL-7B and our AGILE-trained model when responding to general downstream tasks. We observe that **Qwen2.5-VL-7B exhibits dispersed and inconsistent attention**, especially in high-resolution scenarios or tasks requiring fine-grained visual cues. This lack of focus on critical visual evidence limits both its perceptual precision and reasoning effectiveness. In contrast, the AGILE-trained model consistently allocates **highly concentrated attention to key visual tokens**, enabling more accurate extraction of fine details and more reliable visual reasoning in multimodal tasks.
>
> We have incorporated the visualization results and corresponding analyses into **Figure 12** and **Appendix G** of the revised manuscript, which more clearly and substantively demonstrate that AGILE leads to authentic improvements in perceptual and reasoning capabilities.
>
> ***********

---

> > ### Author Response · Authors · 2025-11-19
> > **Detailed Rebuttal (2/3)**
> >
> > > **W2:** The reliance on python code generation for performing actions can limit the model capabilities for further generalization to unstructured tasks. The authors do not mention/explore the use of more traditional tool calling methods to directly call environment APIs instead of generating python code to execute the APIs.
> >
> > When designing the AGILE framework, we have explicitly considered the possibility that VLMs may struggle with open-ended code generation. To minimize this difficulty, we adopt a **predefined Python API** that includes only four primitive operations-**swap**, **observe**, **crop**, and **zoom**. Consequently, the model is not required to produce complete executable code; instead, it only needs to generate the API name along with the corresponding execution parameters. This greatly reduces the complexity of code generation and ensures stable interaction with the environment. Moreover, our experimental results (Table 3: *Main Results*) show that AGILE-trained models exhibits strong generalization even on **unstructured downstream tasks** such as RealWorldQA and HRBench. This indicates that the design of generating Python API calls does not hinder, and may in fact, facilitate broader perceptual and reasoning improvements.
> >
> > To further address this concern, we add experiments using an XML-based tool-calling format. The snippet below shows the differences between our Python-based tool interface and a traditional XML-style tool specification, while also confirming that their complexity levels are comparable.
> >
> > ```
> > 1. Crop a region from an image using normalized to assist in inference:
> > Python:
> > <code>
> > crop_box = [x1, y1, x2, y2]
> > crop_image_{id} = crop(image_id, crop_box)
> > </code>
> >
> > XML:
> > <function_call>
> >     <name>crop_image</name>
> >     <parameters>
> >         <image>image_id</image>
> >         <bbox>[x1, y1, x2, y2]</bbox>
> >     </parameters>
> > </function_call>
> >
> > 2. Zoom in an image by a specified factor (e.g., 1.5 * zoom means 150% zoom).
> > Python:
> > <code>
> > zoom_factor = 1.5
> > zoom_image_{id} = zoom(image_id, zoom_factor)
> > </code>
> >
> > XML:
> > <function_call>
> >     <name>zoom_image</name>
> >     <parameters>
> >         <iamge>image_id</image>
> >         <zoom_factor>factor</zoom_factor>
> >     </parameters>
> > </function_call>
> >
> > 3. Swap_tiles (e.g., top left and bottom left) and call observation function to see the jigsaw progress.
> > Python:
> > <code>
> > state = ["A", "B", "C", "D"]
> > # Swap tiles A (top left) and B (top right)
> > state[0], state[1] = state[1], state[0]
> > # Call observation function to see the result
> > observation_image_{id} = observation(state)
> > </code>
> >
> > XML:
> > <function_call>
> >     <name>swap</name>
> >     <parameters>
> >         <pieces1>state[0]</pieces1>
> >         <pieces2>state[1]</pieces2>
> >     </parameters>
> > </function_call>
> >
> > <function_call>
> >     <name>observation</name>
> >     <parameters>
> >         <jigsaw_state>state</jigsaw_state>
> >     </parameters>
> > </function_call>
> > ```
> >
> > We compare models trained within the AGILE framework using Python code and XML-style tool calls across nine major benchmarks. As shown in the table below, the performance differences between the two formats are minimal, further demonstrating that the choice of Python APIs does not introduce format-specific bias or hinder generalization.
> >
> >
> > |               | MME-RW | RWQA | HRB4K | HRB8K | VStar | MMVP | BLINK | HalBench | MMMU | Avg. |
> > | :-----------: | :----: | :--: | :---: | :---: | :---: | :--: | :---: | :------: | :--: | :--: |
> > | Qwen2.5-VL-7B |  44.6  | 68.5 | 68.8  | 65.3  | 76.4  | 74.3 | 56.4  |   50.1   | 54.8 | 62.1 |
> > |    Python     |  48.4  | 70.2 | 73.0  | 70.5  | 80.6  | 78.0 | 58.0  |   51.9   | 55.8 | 65.2 |
> > |      XML      |  50.4  | 71.1 | 72.3  | 69.8  | 78.5  | 75.3 | 56.7  |   54.0   | 55.4 | 64.8 |
> >
> > ***********
> >
> >
> >
> > > **Q1:** Since the model generates Python code to perform jigsaw actions, how are code generation errors handled during RL training? Are invalid executions penalized, or filtered out before reward calculation?
> >
> > We thank the reviewer for raising this important question. We clarify that within AGILE, invalid executions are **penalized rather than filtered out**. The Python executor returns the corresponding error messages to the model, allowing it to continue the rollout based on this feedback. This mechanism is closely tied to our **reward design**: the reward function includes a **step-penalty term**, so any unnecessary interactions—including those caused by code-generation errors—naturally increase the total number of steps, thereby automatically **reducing the final reward**. The ablation experiments in our response to **Q2** also demonstrate that this mechanism is effective and stable in practice. We have clarified this question more explicitly in **Appendix H**.
> >
> > ***********

---

> > > ### Author Response · Authors · 2025-11-19
> > > **Detailed Rebuttal (3/3)**
> > >
> > > > **Q2:** The paper defines three components (accuracy, format, and step rewards). Could the authors clarify how sensitive overall performance is to the weighting coefficients (α, β, γ)? Have ablation tests been performed to determine the optimal balance among these terms?
> > >
> > > We conducted a series of ablation studies to analyze how sensitive the reward-weighting coefficients are, and the results are shown in the table below.
> > >
> > > |                         | MME-RW | RWQA | HRB4K | HRB8K | VStar | MMVP | BLINK | HalBench | MMMU | Avg. |
> > > | :---------------------: | :----: | :--: | :---: | :---: | :---: | :--: | :---: | :------: | :--: | :--: |
> > > |      Qwen2.5-VL-7B      |  44.6  | 68.5 | 68.8  | 65.3  | 76.4  | 74.3 | 56.4  |   50.1   | 54.8 | 62.1 |
> > > | α, β, γ = 0.8, 0.2, 1.0 |  48.4  | 70.2 | 73.0  | 70.5  | 80.6  | 78.0 | 58.0  |   51.9   | 55.8 | 65.2 |
> > > | α, β, γ = 0.8, 0.2, 0.5 |  49.2  | 69.7 | 72.8  | 69.8  | 78.5  | 79.0 | 57.5  |   50.8   | 55.8 | 64.8 |
> > > | α, β, γ = 0.8, 0.2, 0.0 |  47.4  | 71.1 | 72.4  | 69.8  | 78.5  | 75.3 | 56.7  |   52.8   | 55.6 | 64.4 |
> > > | α, β, γ = 0.9, 0.1, 1.0 |  48.9  | 70.8 | 73.3  | 69.6  | 79.1  | 78.0 | 57.5  |   51.4   | 55.1 | 64.9 |
> > >
> > > **(1) Step-reward ablations**
> > >
> > > We evaluated several configurations, including: (a) removing the step reward entirely (γ=0), and (b) reducing its coefficient by half (γ=0.5).
> > > Removing the step-reward coefficient leads to a significant performance drop, which aligns with our initial design intuition for AGILE. Once the model has acquired a certain level of capability, the step-penalty encourages it to shorten overly long interaction sequences, as excessively long rollouts often indicate sparse or unfocused reasoning. This ultimately enables the model to achieve **sufficient single-step reasoning**. Overall, the results validate that the step-penalty is both beneficial and indispensable for guiding effective learning.
> > >
> > > **(2) Sensitivity to α, β, and γ**
> > >
> > > Adjusting the weighting coefficients causes only minor changes in overall performance, indicating that AGILE is **robust to the selection of α and β**.
> > >
> > > We have added a detailed discussion of the sensitivity to the reward coefficients in **Appendix I.1** of the revised manuscript.
> > >
> > > ***********
> > >
> > > > **Q3:** The model shows improvement across nine benchmarks. Could the authors elaborate on which specific reasoning categories (e.g., fine-grained recognition vs. scene reasoning) benefit most from AGILE's training?
> > >
> > > We appreciate this interesting question and provide further clarification.
> > >
> > > Through analyses of downstream benchmarks, attention-map visualizations, and jigsaw behavior case studies, we identified several categories of tasks that benefit most significantly from AGILE:
> > >
> > > **(1) Fine-grained recognition**
> > >
> > > These tasks require high precision in identifying subtle visual cues, such as detecting small objects, recognizing text, or distinguishing fine-grained visual differences.  AGILE enhances the model's ability to focus on the most critical visual elements, improving its sensitivity to subtle but important features and thereby boosting fine-grained recognition performance.
> > >
> > > **(2) Spatial relation reasoning**
> > >
> > > Benchmarks such as VStarBench contain a substantial number of questions involving spatial positional reasoning. AGILE's interaction-driven training helps the model better attend to spatial layouts and contextual relationships within a scene, substantially improving its ability to understand and reason about complex spatial arrangements. This capability transfers effectively to real-world spatial reasoning tasks.
> > >
> > > **(3) Visual-textual integration**
> > >
> > > AGILE also leads to notable improvements in tasks requiring the integration of visual and textual information. Tasks involving reading product details, interpreting signs, or extracting structured information from charts benefit from AGILE's strengthened ability to jointly process textual cues grounded in visual context.
> > >
> > > In **Appendix G**, we provide a clearer explanation of these task categories and their corresponding improvements under AGILE.

---

> > > > ### Author Response · Authors · 2025-11-27
> > > > **Welcome for Further Feedback**
> > > >
> > > > Dear Reviewer JwKt,
> > > >
> > > > We sincerely thank you for the thorough and constructive feedback that helped guide our additional analyses during the rebuttal period. Following your insightful comments on W1, W2, and Q1–Q3, we have conducted a substantial number of new experiments, visual analyses, and ablation studies in our **Detailed Rebuttal (1-3)**. These revisions were incorporated into the updated manuscript and appendix to more clearly address all points you raised:
> > > >
> > > > - **Regarding W1 (nature of improvements / qualitative gains):**
> > > >   We added extensive *attention-map visualizations* on downstream benchmarks. These analyses clearly show that AGILE-trained models exhibit more coherent and focused visual attention, enabling more precise fine-grained perception and more stable multimodal reasoning. These results are included in **Figure 12 and Appendix G**.
> > > > - **Regarding W2 (tool-calling generality):**
> > > >   Beyond our Python-based interface, we implemented a parallel **XML-style tool-calling format**, demonstrating comparable complexity and almost identical downstream performance across nine benchmarks. This verifies that AGILE's improvements are *not format-specific*, nor dependent on Python code generation.
> > > > - **Regarding Q1 (handling of code-generation errors):**
> > > >   We clarified that invalid executions are *penalized rather than filtered*, with error messages returned to the model. This naturally aligns with the step-penalty term and encourages efficient interaction. Details added in **Appendix H**.
> > > > - **Regarding Q2 (sensitivity to reward weighting):**
> > > >   We expanded the reward-coefficient ablations, including scenarios with γ reduced or removed. Results show stability across α and β and confirm the necessity of the step-penalty. Extended discussion appears in **Appendix I.1**.
> > > > - **Regarding Q3 (categories benefiting most):**
> > > >   We added clearer breakdowns showing the strongest gains in
> > > >   (1) fine-grained recognition,
> > > >   (2) spatial-relation reasoning, and
> > > >   (3) visual–textual integration tasks.
> > > >   These analyses are now summarized in **Appendix G**.
> > > >
> > > > We hope that these substantial additions fully address your thoughtful questions and concerns. If there are any remaining issues that we can clarify further, we would be very grateful to continue the discussion. We kindly ask whether, after reviewing these updates, you might consider revisiting your evaluation or final rating accordingly. Thank you very much again for your time, expertise, and constructive guidance.
> > > >
> > > > Sincerely,
> > > >
> > > > The Authors

---

### Public Comment · ~Junyi_Zhu1 · 2025-11-17
**Link to Jigsaw-R1**

Thank you for sharing this interesting work! I enjoyed reading it and I’m glad to see continuous interest in rule-based RL and structured pretext tasks in multimodal learning.

I’m one of the authors of *Jigsaw-R1: A Study of Rule-based Visual Reinforcement Learning with Jigsaw Puzzles* (Wang et al.; preprint May 2025; accepted to TMLR Oct 2025), which you already cite. Thanks for that. Our paper introduced the use of jigsaw puzzles as a rule-based visual RL pretext task for post-training multimodal LLMs, and studied its impact on generalization and reasoning behaviour. To the best of our knowledge, Jigsaw-R1 was the first to explore this “MLLM + rule-based RL post-training” setting with jigsaw puzzles.

From a reader’s perspective, I believe the relationship between your paper and Jigsaw-R1 could be highlighted a bit more directly. Since both works explore the use of structured visual pretext tasks in RL settings for MLLMs, an explicit experimental comparison (or at least a detailed discussion of why such a comparison is not straightforward) would help the community better appreciate how the approaches complement or diverge from one another.

This is intended merely as a constructive suggestion. I’m happy to see your novel contributions (for instance the adaptation to your modality or environment, and your specific reward and policy engineering), which enriches the broader discussion about how structured visual pretext tasks can enhance the perceptual and reasoning capabilities of multimodal agents under rule-based RL.

---

### Author Response · Authors · 2025-12-03
**Summary of Additional Experiments and Issue Resolutions Provided in the Rebuttal**

In response to the reviewers' constructive suggestions and questions, we have provided detailed clarifications and introduced additional experiments to further strengthen AGILE. These include:
- We conducted a comprehensive set of ablation studies to validate the contribution of each AGILE component.
    - we tested multiple configurations of the reward parameters (α, β, γ) and additionally removed the step‑reward term, showing that the overall reward design is robust and that **step rewards are critical** for improving fine‑grained visual reasoning.
    - our zero‑RL experiments demonstrated that even without expert cold‑start data, **interactive RL alone can yield substantial performance gains**.
    - we removed subsets of the action space (such as cropping/zooming actions, or the full action set) and found that **reducing the structured action space significantly weakens fine‑grained perceptual ability**, although jigsaw‑based RL still provides measurable improvements.
    - we altered the structured representation of the action space by replacing code‑based actions with XML‑based actions, confirming that AGILE's benefits are **not tied to a particular interface** and that different tool‑call formats remain equally effective.
- We added attention visualization and qualitative examples for downstream tasks, showing that AGILE training makes the model more **focused on key visual regions**. We also added analysis of three capability categories: fine-grained recognition, spatial reasoning, and image-text fusion, proving that the performance improvement **comes from the enhancement of real visual and reasoning capabilities**, rather than simply an increase in metrics.
- We added a difficulty scaling experiment, that is, after 2 × 2 training, we further added 3 × 3 jigsaw RL training. The results show that not only does the 3 × 3 capability continue to improve, but the performance of downstream tasks is also further enhanced, showing that AGILE can **scale with task complexity**.
- We evaluated the performance of the AGILE model on external tasks that included novel actions such as rotations, guide lines, and set-of-marks. The model still **performed better in completely unseen toolchains**, demonstrating that the action space is general rather than jigsaw-specific.

---

### Author Response · Authors · 2025-12-03
**Summary for AC Consideration**

Dear Area Chair,

Thank you for your time and for overseeing the review process. We have addressed all comments in detail and provided clarifications based on the valuable suggestions raised during the review process. Below is a brief summary of our rebuttal.

All the reviewers **consistently recognize the novelty and significance of our AGILE framework**, including its scalable task design and controllable difficulty, which effectively **mitigate data‑scarcity challenges** in multimodal learning (Jwkt, ciSx, yrdc); the innovative integration of multimodal models with an interactive agent‑based paradigm, highlighted as a **key step toward autonomous multimodal reasoning** (Jwkt, ciSx, yrdc); the **dramatic leap in jigsaw‑solving performance** (Jwkt, es5M); and the strong **cross‑task generalization** demonstrated across diverse downstream benchmarks (Jwkt, ciSx, es5M, yrdc). We are sincerely grateful for their positive and encouraging feedback.

After carefully reviewing our detailed rebuttal, Reviewer ciSx expressed strong appreciation for the work and raised the score from 6 to 8 (26 Nov 2025, 23:15). Reviewer yrdc also acknowledged a substantial initial misunderstanding of our paper and increased the score from 2 to 4 (25 Nov 2025, 15:17). We note that these score changes occurred before the OpenReview bug that exposed reviewer identities, ensuring that they reflect genuine re‑evaluation rather than external influence.

While we deeply appreciate these re‑assessments, we would also like to raise a few concerns regarding the consistency and fairness of certain parts of the review and discussion process, particularly those led by reviewers es5M and yrdc.
- For Reviewer es5M, many of the core critiques were centered around limited task generality and uncertain sources of gains on general tasks. As we demonstrated in detail through newly added experiments (including larger‑grid training (3 × 3), removal and restructuring of the action space, zero‑RL training without expert demonstrations, and evaluations on tasks with novel tool semantics) each of these concerns was directly addressed and fully resolved by the evidence provided. We invited Reviewer es5M to continue the discussion after these additions, but did not receive further responses. Given that the new experimental results explicitly counter the earlier assumptions, this lack of engagement makes it difficult to understand whether the updated evidence was fully considered.
- For Reviewer yrdc, the initial evaluation appeared to stem from significant misunderstandings of the paper. The issues raised in Q1, Q2, and Q3 were all addressed explicitly in the original submission, and the reviewer's conf‑5 rating seemed inconsistent with the level of engagement with the manuscript. After we provided clarifications and substantial new experiments, the reviewer acknowledged the misunderstanding and stated that the work "looks good." However, despite having no remaining concerns, the score increased only from 2 to 4, which appears inconsistent with both the reviewer's second‑round comments and the standard evaluation criteria applied when misunderstandings have been fully resolved.

We once again thank the Area Chairs and all reviewers for their time and efforts throughout the review process, and we emphasize that we have fully respected the fairness and integrity of the double‑blind reviewing procedure. We sincerely hope that the Area Chair will take into consideration the substantial improvements we have made in response to reviewers' suggestions, as well as the clarifications and extensive new evidence we have provided. We also respectfully ask that you take into account our concerns regarding the reviews led by es5M and yrdc. We remain fully open to any questions or further suggestions you may have, and we sincerely appreciate your efforts in ensuring a fair, rigorous, and balanced evaluation process.

---

### Meta-Review · Area_Chair_PftM · 2026-01-14

**Summary:**

The reviewers raised several points during the review process:

1. Source of the gains and whether there is generalization (3 reviewers). There are concerns about fairness in multi-turn setups, and whether these improvements are actually improvements in visual processing. There is also concern that the improvements come from distillation.
2. Action design space, and whether the code environment limits the capabilities (3 reviewers). There are just a few tools available.
3. The complexity of the experiments (2 reviewers). Since it is a 2x2 jigsaw, the performance may be saturated.
4. Justifications for the rewards and training (3 reviewers). There was questions about the step reward design, format rewards, and their effect on the training.

**Reviewer Concerns:**

The authors did a good job at responding to the reviewer main points. In particular, I found the points 2, 3, 4 to be convincingly answered with more experiments and discussion. I found the answers to 1 to be good, showing that without any trajectory supervision, direct training still yields an improvement. Overall, I also found the paper to be interesting and I believe it will stir interesting discussions at the conference.

**Reviewer Scores:**

I expect the reviewers would have found the answers convincing. The authors did a good job responding to the major points.

---

### Decision · Program_Chairs · 2026-01-26

Accept (Poster)